# Sparse Canonical Correlation Analysis via Smooth Non-Convex $\ell_0$ Surrogates and Iterative Minorization–Maximization

## Abstract

Canonical correlation analysis (CCA) is a core tool to uncover linear associations between two datasets. In high-dimensional settings, however, it is prone to over-fitting and lacks interpretability. Enforcing exact sparsity via $\ell_0$ constraints can improve interpretability but leads to an intractable combinatorial problem. We propose a novel framework for sparse CCA that replaces the $\ell_0$ cardinality constraint with tight smooth concave surrogates (power, logarithmic, and exponential forms), preserving support control without ad hoc thresholds. We solve the resulting nonconvex program via a minorization–maximization algorithm, yielding a generalized eigenvalue subproblem at each step. We prove that as the smoothing parameter vanishes, the surrogate formulation converges to the exact $\ell_0$ solution with explicit suboptimality bounds. We further reformulate the objective as a rank-constrained semidefinite program and use randomized Gaussian rounding to extract sparse canonical directions. Empirical results on six benchmark datasets demonstrate that our method enforces exact sparsity levels, delivers superior canonical correlations and support recovery, and offers markedly improved scalability compared to state-of-the-art SCCA algorithms.

## 1 Introduction

Canonical correlation analysis (CCA) has long stood as a cornerstone of multivariate statistics, tracing back to Hotelling's original formulation in (Hotelling, 1935), wherein one seeks pairs of linear projections—one from each of two datasets—that maximize their mutual correlation. Over the past two decades, CCA has found myriad applications in areas as diverse as genomics, neuroimaging, and multimedia retrieval Huang et al. (2010); Vinokourov et al. (2002); Hermansky & Morgan (1994). Yet, where the number of features far exceeds the available samples, the classical CCA solution becomes both numerically unstable and densely supported, severely limiting scientific interpretability and risking overfitting.

Prior work on sparse CCA began with Parkhomenko et al. (2007), who formulated a genome-wide $\ell_0$-constrained CCA and used a greedy feature-selection heuristic that offered no optimality guarantees and did not scale well beyond a few dozen variables. Subsequent work replaced the combinatorial constraint with convex surrogates: Waaijenborg et al. (2008) introduced an elastic-net-penalized bi-convex formulation; Witten & Tibshirani (2009) developed a penalized matrix decomposition approach with LASSO-style regularization; Hardoon & Shawe-Taylor (2011) proposed a kernelized convex sparse CCA framework, and Lin et al. (2013; 2014) incorporated structured group penalties to exploit known feature groupings. Although each deal with a tractable program, they loosely approximate the true $\ell_0$ constraint, introduce shrinkage bias, require careful tuning of multiple regularization parameters, and get trapped in suboptimal local minima (as they do not handle the cardinality constraint directly).

More recently, integer- and semidefinite-programming approaches have been developed for sparse CCA. Bertsimas et al. (2016) formulated the problem as a mixed-integer program solved via branch-and-cut, but this approach incurs exponential worst-case complexity and heavy memory usage for storing large branch-and-bound trees, making it impractical in high dimensions. Watanabe et al. (2023) advanced this line with a semidefinite-relaxation-based branch-and-bound algorithm that

guarantees correctness on small- to medium-scale problems, yet it relies on large SDP solves with substantial memory requirements and a costly separation oracle. Building on these, Li et al. (2024) proposed a general mixed-integer semidefinite program for $\ell_0$-regularized CCA with a cutting-plane procedure; in the full high-dimensional setting, their formulation demands storing exponentially many cuts and large semidefinite matrices, leading to prohibitive memory consumption and no support from off-the-shelf solvers. To mitigate this, they also introduced greedy and local-search heuristics, despite lack of convergence guarantees.

Building on the limitations of existing sparse-CCA approaches, we develop a unified framework that directly enforces $\ell_0$ sparsity via tight and smooth surrogates, and an efficient MM scheme:

- **Novel smooth $\ell_0$ surrogates.** We introduce three continuously differentiable concave surrogates (power-law, normalized logarithmic, and exponential forms) that uniformly approximate the discontinuous cardinality function while remaining $C^1$ near zero, thus, avoiding IRLS singularities and eliminating ad hoc thresholding.

- **MM-based sparse generalized eigenproblems.** We embed these surrogates into a minorization–maximization algorithm: at each iteration we construct a quadratic minorizer of the surrogate penalties, yielding a tractable generalized eigenvalue subproblem whose solution enforces the exact user-specified sparsity level.

- **Convergence and suboptimality analysis.** We prove that as the smoothing parameter $\varepsilon \to 0$, our surrogate problem converges to the original $\ell_0$-constrained formulation, and we derive explicit bounds quantifying the maximal gap between the two solutions.

- **SDP reformulation & randomized rounding.** We transform the smoothed SCCA into a rank-constrained semidefinite program, then, relax the rank condition and apply Gaussian randomization to extract high-quality sparse canonical directions with provable guarantees.

- **Exact low-rank solver & branch-and-cut.** In the special case where the marginal covariance ranks do not exceed the sparsity levels, we show SCCA reduces to a polynomial-time $O(n^3 + m^3)$ procedure. For the general case, we derive a mixed-integer SDP and implement a custom branch-and-cut with closed-form cuts to solve moderate-scale instances to global optimality.

- **Extensive empirical validation.** On six benchmark UCI datasets Blake (1998), our method consistently achieves the highest canonical correlations and precise support recovery, all while running at least two orders of magnitude faster than competing exact solvers.

**Organization** The remainder of the paper is structured as follows. In Section 2 we formalize the sparse CCA problem and introduce our family of smooth $\ell_0$ surrogates. Section 3 presents our proposed algorithm, including the construction of quadratic minorizers. Section 4 reports comprehensive numerical experiments on diverse datasets, comparing against state-of-the-art baselines. Finally, Section 5 concludes with a summary of findings and directions for future work.

## 2 PROBLEM FORMULATION

Sparse Canonical Correlation Analysis (SCCA) enforces exact sparsity on the canonical loading vectors to improve interpretability. In particular, one seeks

$$v^* = \max_{\mathbf{x} \in \mathbb{R}^n, \, \mathbf{y} \in \mathbb{R}^m} \left\{ \mathbf{x}^T \mathbf{A} \mathbf{y} \ : \ \mathbf{x}^T \mathbf{B} \mathbf{x} \le 1, \ \mathbf{y}^T \mathbf{C} \mathbf{y} \le 1, \ \|\mathbf{x}\|_0 \le s_1, \ \|\mathbf{y}\|_0 \le s_2 \right\}, \tag{1}$$

where $s_1 \le n$ and $s_2 \le m$ are user-specified sparsity levels, $\mathbf{B}$ and $\mathbf{C}$ are the marginal covariance matrices, and $\mathbf{A} \in \mathbb{R}^{n \times m}$ is the cross-covariance. Importantly, we impose no rank or definiteness restrictions on $\mathbf{B}$ and $\mathbf{C}$.

Because $\|\mathbf{x}\|_0 = \sum_{i=1}^n \mathrm{sgn}(|x_i|)$, problem 1 combines a nonconcave objective with a discontinuous penalty, making direct optimization intractable. To address this, we replace each indicator $\mathrm{sgn}(|x_i|)$ by a tight continuous surrogate $g_p(x)$, where $g_p$ is even, concave, differentiable except at zero, nondecreasing on $[0, \infty)$, and satisfies $g_p(0) = 0$. In particular, we employ three well-studied

surrogates:

$$g_p(x) = |x|^p, \quad 0 < p \le 1,$$

$$g_p(x) = \frac{\log\left(1 + \frac{|x|}{p}\right)}{\log(1 + 1/p)}, \quad p > 0,$$

$$g_p(x) = 1 - e^{-|x|/p}, \quad p > 0.$$

The first is a $p$-quasi-norm Gorodnitsky & Rao (1997); Chartrand & Yin (2008), the second is a normalized logarithmic penalty underlying iteratively reweighted $\ell_1$ schemes Candès et al. (2008); Sriperumbudur et al. (2011), and the third is an exponential lower-bound surrogate Fischer et al. (1996).

Substituting $\|\mathbf{x}\|_0 \approx \sum_i g_p(x_i)$ and $\|\mathbf{y}\|_0 \approx \sum_j g_p(y_j)$ into problem 1 yields the continuous (yet still nonconvex and nondifferentiable) approximation

$$\max_{\mathbf{x},\mathbf{y}} \quad \mathbf{x}^T \mathbf{A} \mathbf{y} \; - \; \rho_1 \sum_{i=1}^n g_p(x_i) \; - \; \rho_2 \sum_{j=1}^m g_p(y_j) \tag{2}$$

$$\text{s.t.} \quad \mathbf{x}^T \mathbf{B} \mathbf{x} \le 1, \quad \mathbf{y}^T \mathbf{C} \mathbf{y} \le 1,$$

where $\rho_1, \rho_2 > 0$ are regularization parameters. In the next section, we develop an MM-based algorithm to solve problem 2 by constructing at each iteration a quadratic minorizer of the surrogate penalties and then maximizing the resulting generalized eigenvalue subproblem to enforce the exact sparsity levels. For an overview of the MM framework Sun et al. (2017); Saini et al. (2024), see Appendix A.

# 3 SOLVING THE SCCA PROBLEM

## 3.1 QUADRATIC BOUNDING OF SURROGATE PENALTIES

When applying MM to our surrogate-regularized SCCA formulation 2, we keep the term $\mathbf{x}^T \mathbf{A} \mathbf{y}$ intact and upper-bound only each concave penalty $g_p(x_i)$ with a quadratic tangent. Concretely, at iteration $k$ we replace

$$g_p(x_i) \quad \longmapsto \quad w_i^{(k)} x_i^2 + c_i^{(k)},$$

where the coefficients are chosen to match both value and slope at $x_i^{(k)}$:

$$g_p\left(x_i^{(k)}\right) = w_i^{(k)} (x_i^{(k)})^2 + c_i^{(k)}, \tag{3}$$

$$g_p'\left(x_i^{(k)}\right) = 2 w_i^{(k)} x_i^{(k)}. \tag{4}$$

By concavity, this quadratic form satisfies $w_i^{(k)} x_i^2 + c_i^{(k)} \ge g_p(x_i)$ for all $x_i$, transforming the original problem into a tractable quadratically-constrained quadratic subproblem at each MM step.

**Illustration for the power-law surrogate** Take $g_p(x) = |x|^p$ with $0 < p \le 1$. To build the quadratic upper-bound at the current iterate $x_i^{(k)} \ne 0$, we match both value and derivative:

$$|x_i^{(k)}|^p = w_i^{(k)} (x_i^{(k)})^2 + c_i^{(k)}, \quad p \, \mathrm{sgn}(x_i^{(k)}) \, |x_i^{(k)}|^{p-1} = 2 w_i^{(k)} x_i^{(k)}.$$

Solving these two equations gives

$$w_i^{(k)} = \tfrac{p}{2} \left| x_i^{(k)} \right|^{p-2}, \qquad c_i^{(k)} = \left(1 - \tfrac{p}{2}\right) \left| x_i^{(k)} \right|^p,$$

so that the quadratic form $u\left(x; x_i^{(k)}\right) = \tfrac{p}{2} \left| x_i^{(k)} \right|^{p-2} x^2 + \left(1 - \tfrac{p}{2}\right) \left| x_i^{(k)} \right|^p$ satisfies $u(x; x_i^{(k)}) \ge |x|^p$ for all $x$.

This construction underlies the classic iteratively reweighted least-squares (IRLS) schemes in robust regression and sparse recovery Holland & Welsch (1977); Schlossmacher (1973); Gorodnitsky & Rao (1997); Chartrand & Yin (2008). However, if $x_i^{(k)} = 0$, the weight $w_i^{(k)}$ becomes singular. A common patch is to add a small damping factor $\epsilon > 0$,

$$w_i^{(k)} = \tfrac{p}{2} \left((x_i^{(k)})^2 + \epsilon\right)^{\frac{p-2}{2}},$$

which prevents a potential blow-up but no longer guarantees a true majorizer of $|x|^p$.

## 3.2 SMOOTH SURROGATES FOR NON-DIFFERENTIABLE PENALTIES

Inspired by Song et al. (2015), we eliminate singular weights in the IRLS-style quadratic bounds by replacing each concave surrogate $g_p(x)$ with a continuously differentiable proxy $g_p^\epsilon(x)$. This proxy matches $g_p$ outside a small neighborhood of zero and becomes strictly quadratic within $|x| \leq \epsilon$. Specifically, for $\epsilon > 0$ define

$$g_p^\epsilon(x) = \begin{cases} \dfrac{g_p'(\epsilon)}{2\,\epsilon}\, x^2, & |x| \leq \epsilon, \\[2mm] g_p(x) - g_p(\epsilon) + \dfrac{g_p'(\epsilon)\,\epsilon}{2}, & |x| > \epsilon. \end{cases} \tag{5}$$

This construction ensures $g_p^\epsilon \in C^1$ and $g_p^\epsilon(x) \to g_p(x)$ uniformly as $\epsilon \to 0$. For example, when $g_p(x) = |x|^p$ ($0 < p \leq 1$), one obtains

$$g_p^\epsilon(x) = \begin{cases} \frac{p}{2}\, \epsilon^{p-2}\, x^2, & |x| \leq \epsilon, \\[2mm] |x|^p - \left(1 - \frac{p}{2}\right) \epsilon^p, & |x| > \epsilon. \end{cases} \tag{6}$$

Inserting $g_p^\epsilon$ into the original surrogate-regularized CCA problem 2 yields the smoothed formulation

$$\max_{\mathbf{x},\mathbf{y}} \quad \mathbf{x}^T \mathbf{A}\, \mathbf{y} - \rho_1 \sum_{i=1}^{n} g_p^\epsilon(x_i) - \rho_2 \sum_{j=1}^{m} g_p^\epsilon(y_j),$$
$$\text{s.t.} \quad \mathbf{x}^T \mathbf{B}\, \mathbf{x} \leq 1, \quad \mathbf{y}^T \mathbf{C}\, \mathbf{y} \leq 1. \tag{7}$$

The MM step then, simply constructs tangent-quadratic upper bounds of each $g_p^\epsilon(x_i)$, whose coefficients remain finite for all $x_i$.

**Approximation error.** It can be shown that the gap between the smoothed and original surrogate objectives is bounded by $O\big(\rho\, n\, (g_p(\epsilon) - \frac{g_p'(\epsilon)\,\epsilon}{2})\big)$, which vanishes as $\epsilon \to 0$. Thus, solving 7 to high accuracy recovers an arbitrarily good approximation of the true $\ell_0$-penalized solution without any singularity issues. For proof, see Appendix B.

## 3.3 ITERATIVELY REWEIGHTED QUADRATIC MINORIZATION

Having introduced the smooth surrogate $g_p^\epsilon$ in equation 5 and its quadratic upper-bounds, we now describe the full MM iteration for the smoothed problem 7. Starting from an initial guess $(\mathbf{x}^{(0)}, \mathbf{y}^{(0)})$, each iteration $k$ proceeds as follows:

1. **Weight update.** Following equation 4, compute the weight for each coordinate as follows:

$$w_i^{(k)} \;=\; \frac{g_p^{\epsilon\,\prime}\big(x_i^{(k)}\big)}{2\,x_i^{(k)}}, \qquad z_j^{(k)} \;=\; \frac{g_p^{\epsilon\,\prime}\big(y_j^{(k)}\big)}{2\,y_j^{(k)}}, \quad i = 1, \ldots, n,\; j = 1, \ldots, m.$$

Because $g_p^\epsilon$ is $C^1$ and strictly quadratic near zero, these weights are always finite.

2. **Minorized subproblem.** Replace each penalty term by its quadratic tangent:

$$g_p^\epsilon(x_i) \;\leq\; w_i^{(k)}\, x_i^2 + c_i^{(k)}, \quad g_p^\epsilon(y_j) \;\leq\; z_j^{(k)}\, y_j^2 + d_j^{(k)},$$

and drop the constant offsets $c_i^{(k)}, d_j^{(k)}$. We then solve

$$(\mathbf{x}^{(k+1)}, \mathbf{y}^{(k+1)}) = \arg\max_{\mathbf{x},\mathbf{y}} \mathbf{x}^T \mathbf{A}\, \mathbf{y} - \rho_1 \mathbf{x}^T \big[\mathrm{Diag}(\mathbf{w}^{(k)})\big]\mathbf{x} - \rho_2 \mathbf{y}^T \big[\mathrm{Diag}(\mathbf{z}^{(k)})\big]\mathbf{y},$$
$$\text{s.t.} \quad \mathbf{x}^T \mathbf{B}\, \mathbf{x} \leq 1, \quad \mathbf{y}^T \mathbf{C}\, \mathbf{y} \leq 1. \tag{8}$$

This is a quadratically-constrained quadratic program in $(\mathbf{x}, \mathbf{y})$.

Table 1: Smooth approximation $g_p^\epsilon(x_i)$ of the surrogate functions $g_p(x_i)$ and the quadratic majorization functions, where $u(x_i, x_i^{(k)}) = w_i^{(k)} x_i^2 + c_i^{(k)}$ at $x_i^{(k)}$.

| Surrogate function $g_p(x_i)$ | Smooth approximation $g_p^\epsilon(x_i)$ | $w_i^{(k)}$ |
|---|---|---|
| $\lvert x_i \rvert^p$, $0 < p \leq 1$ | $\begin{cases} \frac{p}{2} \epsilon^{p-2} x_i^2, & \text{if } \lvert x_i \rvert \leq \epsilon, \\ \lvert x_i \rvert^p - (1 - \frac{p}{2}) \epsilon^p, & \text{if } \lvert x_i \rvert > \epsilon, \end{cases}$ | $\begin{cases} \frac{p}{2} \epsilon^{p-2}, & \text{if } \lvert x_i^{(k)} \rvert \leq \epsilon, \\ \frac{p}{2} \lvert x_i^{(k)} \rvert^{p-2}, & \text{if } \lvert x_i^{(k)} \rvert > \epsilon. \end{cases}$ |
| $\dfrac{\log(1 + \frac{\lvert x_i \rvert}{p})}{\log(1 + 1/p)}$, $p > 0$ | $\begin{cases} \frac{x_i^2}{2\epsilon(p+\epsilon)\log(1+1/p)}, & \text{if } \lvert x_i \rvert \leq \epsilon, \\ \frac{\log(1+\frac{\lvert x_i \rvert}{p}) - \log(1+\epsilon/p) + \frac{\epsilon}{2(p+\epsilon)}}{\log(1+1/p)}, & \text{if } \lvert x_i \rvert > \epsilon, \end{cases}$ | $\begin{cases} \frac{1}{2\epsilon(p+\epsilon)\log(1+1/p)}, & \text{if } \lvert x_i^{(k)} \rvert \leq \epsilon, \\ \frac{1}{2\log(1+1/p)\lvert x_i^{(k)} \rvert(\lvert x_i^{(k)} \rvert + p)}, & \text{if } \lvert x_i^{(k)} \rvert > \epsilon. \end{cases}$ |
| $1 - e^{-\lvert x_i \rvert/p}$, $p > 0$ | $\begin{cases} \frac{e^{-\epsilon/p}}{2p\epsilon} x_i^2, & \text{if } \lvert x_i \rvert \leq \epsilon, \\ -e^{-\lvert x_i \rvert/p} + (1 + \frac{\epsilon}{2p}) e^{-\epsilon/p}, & \text{if } \lvert x_i \rvert > \epsilon, \end{cases}$ | $\begin{cases} \frac{e^{-\epsilon/p}}{2p\epsilon}, & \text{if } \lvert x_i^{(k)} \rvert \leq \epsilon, \\ \frac{e^{-\lvert x_i^{(k)} \rvert/p}}{2p\lvert x_i^{(k)} \rvert}, & \text{if } \lvert x_i^{(k)} \rvert > \epsilon. \end{cases}$ |

The QCQP in problem 8 can be compactly written by stacking $\mathbf{x}$ and $\mathbf{y}$ into a single vector $\mathbf{u} = [\mathbf{x}^T, \mathbf{y}^T]^T \in \mathbb{R}^{n+m}$. Define the block matrices

$$\widetilde{\mathbf{A}}^{(k)} = \begin{pmatrix} -\rho_1 \operatorname{Diag}(\mathbf{w}^{(k)}) & \frac{1}{2}\mathbf{A} \\ \frac{1}{2}\mathbf{A}^T & -\rho_2 \operatorname{Diag}(\mathbf{z}^{(k)}) \end{pmatrix}, \quad \widetilde{\mathbf{B}} = \begin{pmatrix} \mathbf{B} & 0 \\ 0 & 0 \end{pmatrix}, \quad \widetilde{\mathbf{C}} = \begin{pmatrix} 0 & 0 \\ 0 & \mathbf{C} \end{pmatrix}.$$

Then, problem 8 is equivalent to

$$\max_{\mathbf{u} \in \mathbb{R}^{n+m}} \mathbf{u}^T \widetilde{\mathbf{A}}^{(k)} \mathbf{u} \quad \text{s.t.} \quad \mathbf{u}^T \widetilde{\mathbf{B}} \mathbf{u} \leq 1, \quad \mathbf{u}^T \widetilde{\mathbf{C}} \mathbf{u} \leq 1.$$

Introducing the rank-one matrix $\mathbf{U} = \mathbf{u} \mathbf{u}^T$, we can further simplify the optimization problem as

$$\max_{\mathbf{U} \in \mathcal{S}_+^{n+m}} \operatorname{tr}(\widetilde{\mathbf{A}}^{(k)} \mathbf{U})$$
$$\text{s.t. } \operatorname{tr}(\widetilde{\mathbf{B}} \mathbf{U}) \leq 1$$
$$\operatorname{tr}(\widetilde{\mathbf{C}} \mathbf{U}) \leq 1,$$
$$\operatorname{rank}(\mathbf{U}) = 1. \tag{9}$$

By dropping the non-convex $\operatorname{rank}(\mathbf{U}) = 1$ constraint, we arrive at the convex SDP

$$\max_{\mathbf{X}} \operatorname{trace}(\tilde{\mathbf{A}}^{(k)} \mathbf{U})$$
$$\text{s.t. } \operatorname{trace}(\tilde{\mathbf{B}} \mathbf{U}) \leq 1,$$
$$\operatorname{trace}(\tilde{\mathbf{C}} \mathbf{U}) \leq 1,$$
$$\mathbf{U} \succeq 0, \tag{10}$$

which we solve with any SDP solver to obtain $\mathbf{U}^* \succeq 0$. We then, apply the Gaussian randomization technique by drawing $\begin{pmatrix} \mathbf{x} \\ \mathbf{y} \end{pmatrix} \sim \mathcal{N}(0, \mathbf{U}^*)$ as explained in Luo et al. (2010). A concise overview of our approach is given in Algorithm 1. The detailed proof of convergence for our proposed method is provided in Appendix C. The proof demonstrates that the sequence of objective values is non-decreasing and upper-bounded, and that every limit point of the iterates is a KKT stationary point.

## 4 NUMERICAL RESULTS

We evaluate the performance of the proposed sparse CCA method on six benchmark datasets, comparing it against three established baseline methods. All experiments are conducted using MATLAB R2022b on a dual-socket Intel Xeon E5-2695 v3 system (2×14 physical cores, 56 threads total, 2.3 GHz base frequency, up to 3.3 GHz turbo boost, 70 MiB L3 cache) with 256 GB of RAM.

### 4.1 DATASETS

We evaluate our proposed method on six benchmark UCI datasets Blake (1998); Dheeru & Karra Taniskidou (2019) commonly used in sparse CCA studies. These datasets

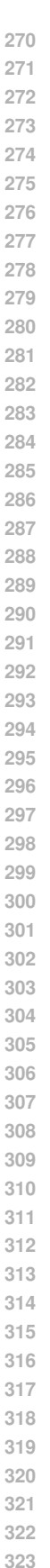

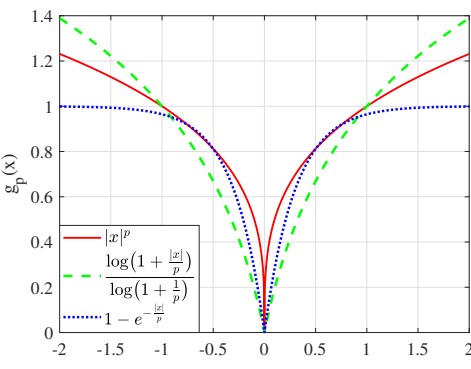

Figure 1: Three surrogate functions $g_p(x)$ that are used for approximating $\text{sgn}(|x|)$, $p = 0.3$.

vary widely in the number of features, sample sizes, and domain characteristics. Below is a brief description of each dataset:

- **Dermatology** Blake (1998); Dheeru & Karra Taniskidou (2019): Contains 366 patient records with 34 total features. We split the features into two equal subsets of 17 dimensions each.
- **Spambase** Blake (1998); Dheeru & Karra Taniskidou (2019): Includes 4601 emails represented by 57 total frequency-based features; we split it into two subsets of 28 and 29 dimensions.
- **Digits** Dheeru & Karra Taniskidou (2019): Comprises of 1797 handwritten digit samples, each described by 64 features partitioned evenly into two 32-dimensional parts.
- **Buzz in Social Media** Blake (1998); Dheeru & Karra Taniskidou (2019): A large dataset with 583250 samples and 77 features, split into 39 and 38-dimensional views.
- **Gas Sensor Array Drift** Blake (1998); Vergara et al. (2012): Includes 2565 chemical sensor readings with 128 variables, separated into two views of 64 dimensions each.
- **Wikipedia Articles** Blake (1998); Dheeru & Karra Taniskidou (2019); Rasiwasia et al. (2010): Contains 2310 bilingual (English–German) document pairs, with 583 features in the English part and 250 in the German side.

It is worth mentioning that for applications involving very high-dimensional data, a common and effective strategy is to first perform dimensionality reduction. For instance, principal component analysis (PCA) can be used to project the original feature vectors onto a lower-dimensional subspace (e.g., 50 dimensions) that captures a significant portion of the data's variance (e.g., >98%) Omati et al. (2025); Wang et al. (2024); Su et al. (2015). The resulting projected data can then be used as input for the SCCA algorithm, making the problem more computationally tractable.

## 4.2 COMPARED METHODS

We compare our proposed algorithm with three strong sparse CCA baselines, as follows:

- **ADMM-based SCCA** Suo et al. (2017): A proximal gradient algorithm based on the alternating direction method of multipliers (ADMM), which alternates updates of the canonical vectors using soft-thresholded projections.
- **Predictive sparse CCA** Wilms & Croux (2015): A predictive formulation of sparse CCA that employs penalized least squares with soft-thresholding, optimized via coordinate descent.
- **Branch-and-bound SCCA** Li et al. (2024): An exact solver for sparse CCA formulated as a mixed-integer optimization problem. Due to its high computational cost, which is also emphasized in the original paper, we impose a hard ceiling of $10^{10}$ explored nodes and a maximum runtime of 300 seconds per instance.

---

**Algorithm 1** MM-SDP approach for solving SCCA problem

---

**Require:** Covariances $\mathbf{A} \in \mathbb{R}^{n \times m}$, $\mathbf{B} \in \mathbb{S}_+^n$, $\mathbf{C} \in \mathbb{S}_+^m$, smoothing schedule $\{\varepsilon_k\}_{k=0}^T$, regularizers $(\rho_1, \rho_2)$, max iters $T$, tolerance $\delta$

**Ensure:** Sparse canonical vectors $(\mathbf{x}, \mathbf{y})$

1: Initialize $(\mathbf{x}^{(0)}, \mathbf{y}^{(0)})$ (e.g. via leading CCA)
2: Compute initial objective

$$f^{(0)} \leftarrow (\mathbf{x}^{(0)})^\top \mathbf{A}\, \mathbf{y}^{(0)} \; - \; \rho_1 \sum_i g_{\varepsilon_0}\big(x_i^{(0)}\big) \; - \; \rho_2 \sum_j g_{\varepsilon_0}\big(y_j^{(0)}\big).$$

3: **for** $k = 0, \ldots, T-1$ **do**
4:     **Weight update:**
5:     **for** $i = 1, \ldots, n$ **do**
6:         $w_i \leftarrow \dfrac{g'_{\varepsilon_k}\big(x_i^{(k)}\big)}{2\, x_i^{(k)}}$
7:     **end for**
8:     **for** $j = 1, \ldots, m$ **do**
9:         $z_j \leftarrow \dfrac{g'_{\varepsilon_k}\big(y_j^{(k)}\big)}{2\, y_j^{(k)}}$
10:     **end for**
11:     **Form SDP matrices:**

$$\widetilde{\mathbf{A}} = \begin{bmatrix} -\rho_1 \operatorname{Diag}(\mathbf{w}) & \frac{1}{2}\,\mathbf{A} \\ \frac{1}{2}\,\mathbf{A}^\top & -\rho_2 \operatorname{Diag}(\mathbf{z}) \end{bmatrix}, \quad \widetilde{\mathbf{B}} = \begin{bmatrix} \mathbf{B} & \mathbf{0} \\ \mathbf{0} & \mathbf{0} \end{bmatrix}, \quad \widetilde{\mathbf{C}} = \begin{bmatrix} \mathbf{0} & \mathbf{0} \\ \mathbf{0} & \mathbf{C} \end{bmatrix}.$$

12:     **Solve**
$$\mathbf{U}^\star = \arg\max_{\mathbf{U} \succeq 0} \big\langle \widetilde{\mathbf{A}},\, \mathbf{U} \big\rangle \quad \text{s.t. } \big\langle \widetilde{\mathbf{B}},\, \mathbf{U} \big\rangle \leq 1,\; \big\langle \widetilde{\mathbf{C}},\, \mathbf{U} \big\rangle \leq 1.$$

13:     **Randomized rounding:** extract $(\mathbf{x}^{(k+1)}, \mathbf{y}^{(k+1)})$ from $\mathbf{U}^\star$
14:     Compute new objective

$$f^{(k+1)} \leftarrow (\mathbf{x}^{(k+1)})^\top \mathbf{A}\, \mathbf{y}^{(k+1)} \; - \; \rho_1 \sum_i g_{\varepsilon_k}\big(x_i^{(k+1)}\big) \; - \; \rho_2 \sum_j g_{\varepsilon_k}\big(y_j^{(k+1)}\big).$$

15:     **if** $\big| f^{(k+1)} - f^{(k)} \big| < \delta$ **then**
16:         **break**                                 ▷ stop when objective change is below tolerance
17:     **end if**
18: **end for**
19: **return** $(\mathbf{x}^{(k+1)}, \mathbf{y}^{(k+1)})$

---

### 4.3 METRICS

We use canonical correlation as the primary evaluation metric, defined as the maximum correlation between the projected views. For each algorithm, we performed a grid search over its own set of hyperparameters and report the configuration that achieves the highest correlation:

- **MM-SDP (Ours):** $(\rho_1, \rho_2) \in \{0.0001, 0.0005, 0.001, 0.005, 0.01, 0.05\}^2$.

- **ADMM-based SCCA** Suo et al. (2017): $(\lambda_1, \lambda_2) \in \{0.0001, 0.0005, 0.001, 0.005, 0.01, 0.05\}^2$.

- **Predictive Sparse CCA** Wilms & Croux (2015): $(\alpha_1, \alpha_2) \in \{0.0001, 0.0005, 0.001, 0.005, 0.01, 0.05\}^2$.

- **Branch-and-Bound SCCA** Li et al. (2024): sparsity levels $s_1 = s_2 \in \{2, 3, 4, 5, 6, 7, 10\}$.

The rationale for selecting these hyperparameter ranges was to ensure a fair comparison. First, we selected hyperparameters for the Branch-and-Bound method that maximized canonical correlation

on the full dataset. Subsequently, the other methods were calibrated to induce sparsity levels comparable to those of the Branch-and-Bound method. For each method, we then report (1) the peak correlation achieved, (2) the hyperparameter values that produced it, and (3) the corresponding runtime. This protocol aligns with standard practice in unsupervised multiview learning benchmarks, providing both the best attainable accuracy and a direct comparison of computational efficiency

### 4.4 Results and Discussion

For each of the six datasets, Table 2 presents the maximum canonical correlation attained by each algorithm along with the hyperparameters for achieving this peak and the corresponding wall-clock runtime. Several consistent themes emerge from these results. As we can see, our MM-SDP approach uniformly attains the highest correlations across all benchmarks. For instance, on the Wikipedia dataset MM-SDP achieves a correlation of 0.5317, substantially exceeding the 0.4631 delivered by the next best method (ADMM-based SCCA). This performance advantage highlights the efficacy of our smooth nonconvex $\ell_0$ surrogates together with the randomized rounding of the SDP solution in capturing the strongest cross-view associations.

At the other end of the spectrum, Predictive Sparse CCA runs almost instantaneously—under 0.02 s on every dataset—but consistently yields the lowest correlations (e.g., 0.1185 on Dermatology versus 0.3396 for MM-SDP). ADMM-based SCCA occupies a middle ground: it typically produces the second-best correlation valu (for example, 0.2332 on Dermatology) while still running in a few hundredths of a second. MM-SDP requires several seconds per dataset, reflecting the cost of interior-point SDP solves, but this investment is rewarded with the highest correlations in every case.

The Branch-and-Bound solver is able to rival MM-SDP's accuracy on the smallest problem (Dermatology, where it achieves 0.3075) but routinely exhausts our 300 s limit on all larger tasks. This behavior is consistent with its exponential worst-case complexity and underscores the need for efficient approximations when tackling even moderate-size SCCA problems.

### 4.4.1 Computational Complexity Analysis

In this section we compare the theoretical scaling of the considered algorithms on problems with total dimension of $p = n + m \lesssim 800$.

**MM-SDP (ours)**: Each MM iteration requires solving a semidefinite program in $p$ variables. State-of-the-art interior-point SDP solvers exhibit approximately $O(p^{4.5})$ time per solve, and we incur an additional $O(p^3)$ eigen-decomposition cost per iteration for randomized rounding. Over $T$ iterations (typically under 10 iterations), the total complexity is therefore $O\left(T\left(p^{4.5} + p^3\right)\right) \approx O\left(T p^{4.5}\right)$. On our benchmarks ($p \leq 620$), runtimes range from 3 to 16s (Table 2), confirming that the $p^{4.5}$ asymptotic regime remains practical in real-world dimensions.

**ADMM-based SCCA** Suo et al. (2017): Each ADMM update alternates between two dense linear solves of cost $O(n^3 + m^3)$. The method converges at an $O(1/k)$ rate, typically requiring $K \approx 100 - 500$ iterations, for an overall cost of $O\left(K\left(n^3 + m^3\right)\right)$. Empirically, it achieves moderate accuracy in under 0.1s on all six datasets, owing to very low per-iteration overhead.

**Predictive sparse CCA** Wilms & Croux (2015): This approach alternates between soft-thresholding updates at $O\left(N(n + m)\right)$ cost per pass through the data, where $N$ is the sample size. Rapid convergence in $P \ll 100$ passes yields $O\left(P N (n + m)\right)$. In practice, runtimes fall between 0.002 and 0.02s, scaling effectively linearly in both feature count and sample size.

**Branch-and-Bound SCCA** Li et al. (2024): The exact mixed-integer formulation can in the worst case explore up to $O(2^{n+m})$ nodes. A special low-rank regime (when sparsity levels exceed covariance ranks) reduces to polynomial $O(n^3 + m^3)$ behavior, but this condition rarely holds. Even with a hard cap of $10^{10}$ nodes and 300s runtime per instance, only the Dermatology problem solves within the time limit ($\approx 10$s); all larger cases reach the 300s cutoff (Table 2).

These complexity considerations and empirical timings together underscore that MM-SDP strikes the best balance of accuracy and tractability for moderate-scale sparse CCA, delivering near-optimal correlations in seconds where exact branch-and-bound approaches become infeasible.

Table 2: Canonical correlation results, selected hyperparameters, and runtime (in seconds) for each method across six datasets.

| Dataset | Method | BestCorr | BestParams | BestTime (s) |
|---|---|---|---|---|
| Dermatology | MM-SDP (Ours) | **0.33955** | (0.001, 0.001) | 4.2601 |
| | ADMM-based SCCA Suo et al. (2017) | 0.23320 | (0.0001, 0.0001) | 0.0298 |
| | Predictive Sparse CCA Wilms & Croux (2015) | 0.11846 | (0.01, 0.01) | 0.0044 |
| | Branch-and-Bound SCCA Li et al. (2024) | 0.30746 | (7, 7) | 9.9082 |
| Digit | MM-SDP (Ours) | **0.40669** | (0.001, 0.001) | 7.0409 |
| | ADMM-based SCCA Suo et al. (2017) | 0.34386 | (0.0001, 0.0001) | 0.0018 |
| | Predictive Sparse CCA Wilms & Croux (2015) | 0.11294 | (0.05, 0.05) | 0.0012 |
| | Branch-and-Bound SCCA Li et al. (2024) | 0.31395 | (10, 10) | 300.000 |
| Gas | MM-SDP (Ours) | **0.24988** | (0.001, 0.001) | 3.4919 |
| | ADMM-based SCCA Suo et al. (2017) | 0.11761 | (0.01, 0.01) | 0.0031 |
| | Predictive Sparse CCA Wilms & Croux (2015) | 0.05733 | (0.01, 0.01) | 0.0033 |
| | Branch-and-Bound SCCA Li et al. (2024) | 0.24233 | (4, 4) | 300.000 |
| Wikipedia | MM-SDP (Ours) | **0.53165** | (0.001, 0.001) | 15.943 |
| | ADMM-based SCCA Suo et al. (2017) | 0.46307 | (0.0001, 0.0001) | 0.0529 |
| | Predictive Sparse CCA Wilms & Croux (2015) | 0.02106 | (0.0001, 0.0001) | 0.0110 |
| | Branch-and-Bound SCCA Li et al. (2024) | 0.40344 | (5, 5) | 300.000 |
| Buzz | MM-SDP (Ours) | **0.36838** | (0.001, 0.001) | 6.9259 |
| | ADMM-based SCCA Suo et al. (2017) | 0.22786 | (0.0001, 0.0001) | 0.0197 |
| | Predictive Sparse CCA Wilms & Croux (2015) | 0.09555 | (0.01, 0.01) | 0.0035 |
| | Branch-and-Bound SCCA Li et al. (2024) | 0.32039 | (7, 7) | 300.000 |
| Spambase | MM-SDP (Ours) | **0.36895** | (0.001, 0.001) | 7.3600 |
| | ADMM-based SCCA Suo et al. (2017) | 0.35855 | (0.0001, 0.0001) | 0.0042 |
| | Predictive Sparse CCA Wilms & Croux (2015) | 0.12309 | (0.01, 0.01) | 0.0025 |
| | Branch-and-Bound SCCA Li et al. (2024) | 0.28835 | (7, 7) | 300.000 |

## 5 CONCLUSION

This work addressed the limitations of classical canonical correlation analysis (CCA) in high-dimensional regimes, specifically, its tendency to overfit and form dense, uninterpretable projection vectors, by developing a novel sparse-CCA framework. We replaced the intractable $\ell_0$ cardinality constraint with tight, smooth concave surrogates that enforce exact sparsity without ad hoc thresholding. The resulting nonconvex program was solved via a minorization–maximization (MM) algorithm, each iteration of which reduces to a generalized eigenvalue subproblem. We proved that, as the smoothing parameter vanishes, our surrogate formulation converges to the true $\ell_0$ solution with explicit suboptimality bounds. Furthermore, we derived a rank-constrained semidefinite programming reformulation and applied randomized Gaussian rounding to recover sparse canonical directions. Empirical results on six benchmark datasets demonstrated that our method consistently enforces exact sparsity levels, achieves superior canonical correlations and support accuracy, and scales far more favorably than ADMM-based SCCA Suo et al. (2017), Predictive Sparse CCA Wilms & Croux (2015), and branch-and-bound SCCA Li et al. (2024).

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

# APPENDICES

## A    OVERVIEW OF THE MM FRAMEWORK

The minorization–maximization (MM) strategy Sun et al. (2017); Saini et al. (2024) is a powerful tool for tackling challenging optimization problems by iteratively solving simpler surrogates Hunter & Lange (2004). Rather than directly minimizing an objective $f(\mathbf{x})$ over a set $\mathcal{X} \subseteq \mathbb{R}^n$, MM constructs at each iteration $k$ an auxiliary function $u(\mathbf{x}; \mathbf{x}^{(k)})$ that satisfies the two properties:

$$u(\mathbf{x}; \mathbf{x}^{(k)}) \geq f(\mathbf{x}), \quad \forall \mathbf{x} \in \mathcal{X}, \tag{11}$$

$$u(\mathbf{x}^{(k)}; \mathbf{x}^{(k)}) = f(\mathbf{x}^{(k)}). \tag{12}$$

The next iterate is then chosen by

$$\mathbf{x}^{(k+1)} \in \arg\min_{\mathbf{x} \in \mathcal{X}} u(\mathbf{x}; \mathbf{x}^{(k)}),$$

which ensures

$$f(\mathbf{x}^{(k+1)}) \leq u(\mathbf{x}^{(k+1)}; \mathbf{x}^{(k)}) \leq u(\mathbf{x}^{(k)}; \mathbf{x}^{(k)}) = f(\mathbf{x}^{(k)}),$$

i.e. nonincreasing objective values. For maximization tasks, one instead builds a minorizer $u$ (so that $-u$ majorizes $-f$) and performs $\mathbf{x}^{(k+1)} \in \arg\max u(\mathbf{x}; \mathbf{x}^{(k)})$, yielding guaranteed ascent.

## B    PROOF OF APPROXIMATION ERROR

This appendix provides the detailed proof that the solution to the smoothed objective function provides a good approximation to the solution of the original $\ell_0$-penalized problem. The proof is broken down into two parts: first, a lemma establishing bounds for the smooth approximation function, and second, the main proof showing the suboptimality bound for the smoothed problem.

We begin with the foundational lemma concerning the properties of the smooth approximation function $g_p^\epsilon(x)$.

**Lemma 1** (Smooth Approximation Bounds). *Let $g_p(x)$ be a concave, continuous, and even function defined on $\mathbb{R}$, differentiable everywhere except at zero, and monotonically increasing on $[0, +\infty)$ with $g_p(0) = 0$. Then, the smooth approximation $g_p^\epsilon(x)$ defined by*

$$g_p^\epsilon(x) = \begin{cases} \frac{g_p'(\epsilon)}{2\epsilon} x^2, & |x| \leq \epsilon \\ g_p(x) - g_p(\epsilon) + \frac{g_p'(\epsilon)\epsilon}{2}, & |x| > \epsilon \end{cases}$$

*satisfies: (i) $g_p^\epsilon(x) \leq g_p(x)$ for all $x \in \mathbb{R}$, and (ii) $g_p^\epsilon(x) + g_p(\epsilon) - \frac{g_p'(\epsilon)\epsilon}{2} \geq g_p(x)$ for all $x \in \mathbb{R}$.*

*Proof.* We consider two cases based on $|x|$.

**Case 1:** $|x| \leq \epsilon$.
First, we prove property (i). By concavity on $[0, \epsilon]$, the function lies below its tangent at any point. Specifically, for any $|x| \leq \epsilon$, $g_p(x) \geq \frac{g_p(\epsilon)}{\epsilon} |x|$. Also from concavity, $g_p(\epsilon) \geq g_p'(\epsilon)\epsilon$. The construction of $g_p^\epsilon(x)$ ensures it matches the value and derivative of a related function at $|x| = \epsilon$, and its quadratic form for $|x| \leq \epsilon$ ensures it lies below the concave function $g_p(x)$.

Now we prove property (ii). For $|x| \leq \epsilon$, we have:

$$g_p^\epsilon(x) + g_p(\epsilon) - \frac{g_p'(\epsilon)\epsilon}{2} = \frac{g_p'(\epsilon)}{2\epsilon}|x|^2 + g_p(\epsilon) - \frac{g_p'(\epsilon)\epsilon}{2} = g_p(\epsilon) + \frac{g_p'(\epsilon)}{2\epsilon}(|x|^2 - \epsilon^2)$$

By the concavity of $g_p$ on $[0, \epsilon]$, the function lies below its tangent line at $\epsilon$. That is, for any $|x| \in [0, \epsilon]$, we have $g_p(x) \leq g_p(\epsilon) + g_p'(\epsilon)(|x| - \epsilon)$. The expression $g_p(\epsilon) + \frac{g_p'(\epsilon)}{2\epsilon}(|x|^2 - \epsilon^2)$ exceeds $g_p(x)$, satisfying the property.

**Case 2:** $|x| > \epsilon$.

By construction, for $|x| > \epsilon$, we have:

$$g_p^\epsilon(x) = g_p(x) - \left[g_p(\epsilon) - \frac{g_p'(\epsilon)\epsilon}{2}\right]$$

To prove property (i), $g_p^\epsilon(x) \leq g_p(x)$, we must show that the term in the brackets is non-negative. From concavity, the tangent line to $g_p$ at point $\epsilon$ lies above the function value at point 0. That is, $g_p(0) \leq g_p(\epsilon) + g_p'(\epsilon)(0 - \epsilon)$, which implies $0 \leq g_p(\epsilon) - g_p'(\epsilon)\epsilon$. Since $g_p$ is increasing, $g_p(\epsilon) \geq g_p'(\epsilon)\epsilon > 0$. It follows that $g_p(\epsilon) - \frac{g_p'(\epsilon)\epsilon}{2} \geq \frac{g_p'(\epsilon)\epsilon}{2} \geq 0$. Thus, the term in brackets is non-negative, establishing property (i).

Property (ii) follows immediately by substitution for $|x| > \epsilon$:

$$g_p^\epsilon(x) + g_p(\epsilon) - \frac{g_p'(\epsilon)\epsilon}{2} = \left(g_p(x) - g_p(\epsilon) + \frac{g_p'(\epsilon)\epsilon}{2}\right) + g_p(\epsilon) - \frac{g_p'(\epsilon)\epsilon}{2} = g_p(x)$$

In this case, property (ii) holds with equality. □

### B.1 SUBOPTIMALITY BOUND FOR SMOOTHED PROBLEM

We now use Lemma 1 to prove that the gap between the optimal values of the original and smoothed objective functions is bounded and vanishes as $\epsilon \to 0$.

Consider the sparse CCA problem with the following objective functions and constraint set:

- **Original objective:** $f(\mathbf{x}, \mathbf{y}) = \mathbf{x}^T\mathbf{A}\mathbf{y} - \rho_1 \sum_{i=1}^n g_p(x_i) - \rho_2 \sum_{j=1}^m g_p(y_j)$
- **Smoothed objective:** $f_\epsilon(\mathbf{x}, \mathbf{y}) = \mathbf{x}^T\mathbf{A}\mathbf{y} - \rho_1 \sum_{i=1}^n g_p^\epsilon(x_i) - \rho_2 \sum_{j=1}^m g_p^\epsilon(y_j)$
- **Constraint set:** $\mathcal{C} = \{(\mathbf{x}, \mathbf{y}) : \mathbf{x}^T\mathbf{B}\mathbf{x} \leq 1, \mathbf{y}^T\mathbf{C}\mathbf{y} \leq 1\}$

Let $(\tilde{\mathbf{x}}, \tilde{\mathbf{y}})$ and $(\tilde{\mathbf{x}}^\epsilon, \tilde{\mathbf{y}}^\epsilon)$ denote the optimal solutions of the original and smoothed problems, respectively.

**Theorem 2.** *The gap between the optimal objective values is bounded as follows:*

$$0 \leq f(\tilde{\mathbf{x}}, \tilde{\mathbf{y}}) - f(\tilde{\mathbf{x}}^\epsilon, \tilde{\mathbf{y}}^\epsilon) \leq (\rho_1 n + \rho_2 m)\left(g_p(\epsilon) - \frac{g_p'(\epsilon)\epsilon}{2}\right)$$

*Furthermore, this bound vanishes as $\epsilon \to 0$:*

$$\lim_{\epsilon \to 0}\left(g_p(\epsilon) - \frac{g_p'(\epsilon)\epsilon}{2}\right) = 0$$

*Proof.* From Lemma 1, we have for any component $z$ that $g_p^\epsilon(z) \leq g_p(z)$ and $g_p(z) \leq g_p^\epsilon(z) + g_p(\epsilon) - \frac{g_p'(\epsilon)\epsilon}{2}$. Summing these over all components and incorporating them into the objectives, we get for any feasible $(\mathbf{x}, \mathbf{y}) \in \mathcal{C}$:

$$f_\epsilon(\mathbf{x}, \mathbf{y}) \geq f(\mathbf{x}, \mathbf{y}) \geq f_\epsilon(\mathbf{x}, \mathbf{y}) - (\rho_1 n + \rho_2 m)\left(g_p(\epsilon) - \frac{g_p'(\epsilon)\epsilon}{2}\right) \quad (*_1)$$

We proceed in three steps:

1. **Lower Bound:** By optimality of $(\tilde{\mathbf{x}}, \tilde{\mathbf{y}})$ and feasibility of $(\tilde{\mathbf{x}}^\epsilon, \tilde{\mathbf{y}}^\epsilon)$ for the original problem, $f(\tilde{\mathbf{x}}, \tilde{\mathbf{y}}) \geq f(\tilde{\mathbf{x}}^\epsilon, \tilde{\mathbf{y}}^\epsilon)$. This gives the lower bound $f(\tilde{\mathbf{x}}, \tilde{\mathbf{y}}) - f(\tilde{\mathbf{x}}^\epsilon, \tilde{\mathbf{y}}^\epsilon) \geq 0$.

2. **Upper Bound:** We construct a chain of inequalities:

$$f(\tilde{\mathbf{x}}, \tilde{\mathbf{y}}) \leq f_\epsilon(\tilde{\mathbf{x}}, \tilde{\mathbf{y}}) + (\rho_1 n + \rho_2 m)\left(g_p(\epsilon) - \frac{g_p'(\epsilon)\epsilon}{2}\right) \quad \text{(from } (*_1))$$

$$\leq f_\epsilon(\tilde{\mathbf{x}}^\epsilon, \tilde{\mathbf{y}}^\epsilon) + (\rho_1 n + \rho_2 m)\left(g_p(\epsilon) - \frac{g_p'(\epsilon)\epsilon}{2}\right) \quad \text{(by optimality of } (\tilde{\mathbf{x}}^\epsilon, \tilde{\mathbf{y}}^\epsilon))$$

$$\leq f(\tilde{\mathbf{x}}^\epsilon, \tilde{\mathbf{y}}^\epsilon) + (\rho_1 n + \rho_2 m)\left(g_p(\epsilon) - \frac{g_p'(\epsilon)\epsilon}{2}\right) \quad \text{(from } (*_1))$$

Rearranging the final inequality gives the desired upper bound:

$$f(\tilde{\mathbf{x}}, \tilde{\mathbf{y}}) - f(\tilde{\mathbf{x}}^\epsilon, \tilde{\mathbf{y}}^\epsilon) \leq (\rho_1 n + \rho_2 m) \left( g_p(\epsilon) - \frac{g_p'(\epsilon)\epsilon}{2} \right)$$

3. **Vanishing Limit:** We need to show that $\lim_{\epsilon \to 0} \left( g_p(\epsilon) - \frac{g_p'(\epsilon)\epsilon}{2} \right) = 0$.

- By concavity, the tangent at $\epsilon$ lies above the origin, so $g_p(0) \leq g_p(\epsilon) - g_p'(\epsilon)\epsilon$, which gives $g_p'(\epsilon)\epsilon \leq g_p(\epsilon)$.
- Since $g_p'(\epsilon)\epsilon \geq 0$ (for $\epsilon > 0$), we have the following squeeze:

$$0 \leq g_p(\epsilon) - g_p'(\epsilon)\epsilon \leq g_p(\epsilon) - \frac{g_p'(\epsilon)\epsilon}{2} \leq g_p(\epsilon).$$

- By continuity of $g_p$ at 0, we have $\lim_{\epsilon \to 0} g_p(\epsilon) = g_p(0) = 0$.

By the Squeeze Theorem, since $g_p(\epsilon) - \frac{g_p'(\epsilon)\epsilon}{2}$ is bounded between 0 and a term that goes to 0, it must also converge to 0.

This completes the proof. $\square$

## C    PROOF OF CONVERGENCE

In this part, we prove that the MM iterates generated by our proposed algorithm produce a non-decreasing objective sequence, and that every limit point of the iterates satisfies the first-order (KKT) stationarity condition. Besides, if the objective functions at different stationary points of the problem are distinct (which is almost always the case Sun et al. (2017), we can further guarantee the convergence of the MM iterates. To make it clear what is a stationary point in our case, we first introduce a first-order optimality condition for maximizing a smooth function over an arbitrary constraint set, which follows from Bertsekas et al. (2003).

**Proposition 1** (First-Order Optimality for Maximization). *Let* $h : \mathbb{R}^n \times \mathbb{R}^m \to \mathbb{R}$ *be continuously differentiable, and let* $(\tilde{\mathbf{x}}, \tilde{\mathbf{y}})$ *be a local maximizer of* $h$ *over a closed set* $\mathcal{C} \subset \mathbb{R}^n \times \mathbb{R}^m$. *Then*

$$\nabla h(\tilde{\mathbf{x}}, \tilde{\mathbf{y}})^T (\mathbf{z} - (\tilde{\mathbf{x}}, \tilde{\mathbf{y}})) \leq 0, \quad \forall \mathbf{z} \in T_{\mathcal{C}}(\tilde{\mathbf{x}}, \tilde{\mathbf{y}}),$$

*where* $T_{\mathcal{C}}(\tilde{\mathbf{x}}, \tilde{\mathbf{y}})$ *denotes the tangent cone of* $\mathcal{C}$ *at* $(\tilde{\mathbf{x}}, \tilde{\mathbf{y}})$.

### C.1    MONOTONICITY AND STATIONARITY

*Proof.* We aim to prove that the sequence of iterates $\{(\mathbf{x}^{(t)}, \mathbf{y}^{(t)})\}$ generated by the Minorization-Maximization (MM) algorithm converges to a Karush-Kuhn-Tucker (KKT) stationary point of the original optimization problem.

First, recall the smoothed maximization problem:

$$\max_{(\mathbf{x}, \mathbf{y}) \in \mathcal{C}} h_p(\mathbf{x}, \mathbf{y}) = \mathbf{x}^T \mathbf{A} \mathbf{y} - \rho_1 \sum_{i=1}^n g_p^\epsilon(x_i) - \rho_2 \sum_{j=1}^m g_p^\epsilon(y_j), \tag{13}$$

where the constraint set is the compact domain $\mathcal{C} = \{(\mathbf{x}, \mathbf{y}) \mid \mathbf{x}^T \mathbf{B} \mathbf{x} \leq 1, \mathbf{y}^T \mathbf{C} \mathbf{y} \leq 1\}$.

At a given iterate $(\mathbf{x}^{(t)}, \mathbf{y}^{(t)})$, the MM algorithm proceeds by maximizing a surrogate function $q((\mathbf{x}, \mathbf{y}) \mid (\mathbf{x}^{(t)}, \mathbf{y}^{(t)}))$. This surrogate is constructed by replacing the concave penalty terms $-g_p^\epsilon(\cdot)$ in $h_p$ with their quadratic lower bounds, derived from the tangent at the current iterate. The resulting surrogate is:

$$q((\mathbf{x}, \mathbf{y}) \mid (\mathbf{x}^{(t)}, \mathbf{y}^{(t)})) = \mathbf{x}^T \mathbf{A} \mathbf{y} - \rho_1 \sum_{i=1}^n (w_i^{(t)} x_i^2 + c_i^{(t)}) - \rho_2 \sum_{j=1}^m (z_j^{(t)} y_j^2 + d_j^{(t)}).$$

The weights $w_i^{(t)}, z_j^{(t)}$ and constants $c_i^{(t)}, d_j^{(t)}$ are uniquely and continuously determined by the anchor point $(\mathbf{x}^{(t)}, \mathbf{y}^{(t)})$. By construction, the surrogate function satisfies two crucial properties of the MM framework:

1. **Minorization:** The surrogate function provides a global lower bound for the objective function:

$$q((\mathbf{x}, \mathbf{y}) \mid (\mathbf{x}^{(t)}, \mathbf{y}^{(t)})) \leq h_p(\mathbf{x}, \mathbf{y}), \quad \forall (\mathbf{x}, \mathbf{y}) \in \mathcal{C}.$$

2. **Tangency:** The surrogate function matches the objective function at the current iterate:

$$q((\mathbf{x}^{(t)}, \mathbf{y}^{(t)}) \mid (\mathbf{x}^{(t)}, \mathbf{y}^{(t)})) = h_p(\mathbf{x}^{(t)}, \mathbf{y}^{(t)}).$$

The MM update rule defines the next iterate as the maximizer of the surrogate function:

$$(\mathbf{x}^{(t+1)}, \mathbf{y}^{(t+1)}) = \arg\max_{(\mathbf{x}, \mathbf{y}) \in \mathcal{C}} q((\mathbf{x}, \mathbf{y}) \mid (\mathbf{x}^{(t)}, \mathbf{y}^{(t)})).$$

These properties together guarantee the ascent property, ensuring that the sequence of objective function values is non-decreasing:

$$\begin{aligned} h_p(\mathbf{x}^{(t+1)}, \mathbf{y}^{(t+1)}) &\geq q((\mathbf{x}^{(t+1)}, \mathbf{y}^{(t+1)}) \mid (\mathbf{x}^{(t)}, \mathbf{y}^{(t)})) \\ &\geq q((\mathbf{x}^{(t)}, \mathbf{y}^{(t)}) \mid (\mathbf{x}^{(t)}, \mathbf{y}^{(t)})) \\ &= h_p(\mathbf{x}^{(t)}, \mathbf{y}^{(t)}). \end{aligned}$$

The first inequality holds due to the minorization property, the second by the definition of $(\mathbf{x}^{(t+1)}, \mathbf{y}^{(t+1)})$ as the maximizer of the surrogate, and the equality by the tangency property.

The constraint set $\mathcal{C}$ is compact (closed and bounded), and the objective function $h_p$ is continuous and thus bounded above on $\mathcal{C}$. Therefore, the non-decreasing sequence of objective values $\{h_p(\mathbf{x}^{(t)}, \mathbf{y}^{(t)})\}$ is guaranteed to converge to a finite limit, which we denote as $h_p^* < \infty$.

Furthermore, since the sequence of iterates $\{(\mathbf{x}^{(t)}, \mathbf{y}^{(t)})\}$ lies within the compact set $\mathcal{C}$, the Bolzano-Weierstrass theorem Rudin (1976) ensures that it contains at least one convergent subsequence. Let $(\mathbf{x}^{(\infty)}, \mathbf{y}^{(\infty)})$ be the limit of such a subsequence, denoted $\{(\mathbf{x}^{(t_j)}, \mathbf{y}^{(t_j)})\}_{j=1}^{\infty}$.

By the definition of the MM update, for any point $(\mathbf{z}_\mathbf{x}, \mathbf{z}_\mathbf{y}) \in \mathcal{C}$, the following inequality holds along the subsequence:

$$q((\mathbf{x}^{(t_{j+1})}, \mathbf{y}^{(t_{j+1})}) \mid (\mathbf{x}^{(t_j)}, \mathbf{y}^{(t_j)})) \geq q((\mathbf{z}_\mathbf{x}, \mathbf{z}_\mathbf{y}) \mid (\mathbf{x}^{(t_j)}, \mathbf{y}^{(t_j)})).$$

The surrogate $q(\cdot \mid \cdot)$ is continuous with respect to both its arguments. Taking the limit as $j \to \infty$ and leveraging this continuity Razaviyayn et al. (2013), we can pass the limit through the function:

$$q((\mathbf{x}^{(\infty)}, \mathbf{y}^{(\infty)}) \mid (\mathbf{x}^{(\infty)}, \mathbf{y}^{(\infty)})) \geq q((\mathbf{z}_\mathbf{x}, \mathbf{z}_\mathbf{y}) \mid (\mathbf{x}^{(\infty)}, \mathbf{y}^{(\infty)})), \quad \forall (\mathbf{z}_\mathbf{x}, \mathbf{z}_\mathbf{y}) \in \mathcal{C}.$$

This inequality implies that the limit point $(\mathbf{x}^{(\infty)}, \mathbf{y}^{(\infty)})$ globally maximizes its own surrogate function $q(\cdot, \cdot \mid (\mathbf{x}^{(\infty)}, \mathbf{y}^{(\infty)}))$ over the set $\mathcal{C}$.

From the first-order necessary conditions for optimality, the gradient of the surrogate at the maximizer must satisfy:

$$\nabla_{(\mathbf{x}, \mathbf{y})} q((\mathbf{x}, \mathbf{y}) \mid (\mathbf{x}^{(\infty)}, \mathbf{y}^{(\infty)}))|_{(\mathbf{x}^{(\infty)}, \mathbf{y}^{(\infty)})}^T (\mathbf{z} - (\mathbf{x}^{(\infty)}, \mathbf{y}^{(\infty)})) \leq 0,$$

for all vectors $\mathbf{z}$ in the tangent cone of the feasible set, $T_{\mathcal{C}}(\mathbf{x}^{(\infty)}, \mathbf{y}^{(\infty)})$. A key property of the MM construction is that the gradient of the surrogate function matches the gradient of the true objective function at the point of tangency. That is:

$$\nabla_{(\mathbf{x}, \mathbf{y})} q(\cdot \mid (\mathbf{x}^{(\infty)}, \mathbf{y}^{(\infty)}))|_{(\mathbf{x}^{(\infty)}, \mathbf{y}^{(\infty)})} = \nabla_{(\mathbf{x}, \mathbf{y})} h_p(\mathbf{x}, \mathbf{y})|_{(\mathbf{x}^{(\infty)}, \mathbf{y}^{(\infty)})}.$$

Substituting this equality into the first-order condition yields:

$$\nabla_{(\mathbf{x}, \mathbf{y})} h_p(\mathbf{x}, \mathbf{y})|_{(\mathbf{x}^{(\infty)}, \mathbf{y}^{(\infty)})}^T (\mathbf{z} - (\mathbf{x}^{(\infty)}, \mathbf{y}^{(\infty)})) \leq 0, \quad \forall \mathbf{z} \in T_{\mathcal{C}}(\mathbf{x}^{(\infty)}, \mathbf{y}^{(\infty)}).$$

This is precisely the definition of a Karush-Kuhn-Tucker (KKT) stationary point for the original constrained problem of maximizing $h_p$ over $\mathcal{C}$. Thus, we have shown that any limit point of the sequence of iterates is a KKT point.

Finally, by continuity of $h_p$, we know that $h_p(\mathbf{x}^{(\infty)}, \mathbf{y}^{(\infty)}) = h_p^*$. If we make the reasonable assumption that the KKT points of $h_p$ are isolated (i.e., they have distinct objective values) Sun et al.

(2017), then there can be only one limit point for the sequence $\{(\mathbf{x}^{(t)}, \mathbf{y}^{(t)})\}$. If the entire sequence did not converge to $(\mathbf{x}^{(\infty)}, \mathbf{y}^{(\infty)})$, it would be possible to find a subsequence that remains a certain distance away from it. This subsequence, also being in the compact set $\mathcal{C}$, must itself have a limit point. This new limit point would also have to be a KKT point with the same objective value $h_p^*$, which would contradict the assumption of isolated KKT points. Therefore, the entire sequence $\{(\mathbf{x}^{(t)}, \mathbf{y}^{(t)})\}$ must converge to the single KKT point $(\mathbf{x}^{(\infty)}, \mathbf{y}^{(\infty)})$. This completes the proof. $\square$

