# OpenReview forum: "Sparse Canonical Correlation Analysis via Smooth Non-Convex $\ell_{0}$ Surrogates and Iterative Minorization–Maximization"
_ICLR.cc/2026/Conference — Submitted to ICLR 2026_

### Official Review · Reviewer_FXaa · 2025-10-29

**Soundness:** 2
**Presentation:** 3
**Contribution:** 2
**Rating:** 4
**Confidence:** 3

**Summary:**

The paper proposed MM-SDP for sparse classical canonical correlation analysis (SCCA). Smooth concave surrogates are employed to replace the l0-cardinality constraint. The convergence of the method is also proved.

**Strengths:**

- The proposed method achieves the best performance throughout all the 6 datasets compared to the baselines. The maximum canonical correlation is improved by 0.038 on average compared to the second-best method.
- Taking into account both maximum canonical correlation and runtime, the proposed MM-SDP achieves the best performance.

**Weaknesses:**

- The baselines and related works are limited. Two of the three adopted baselines are proposed over 8 years ago. Moreover, no related works are discussed.
- The analysis of the experimental results is not thorough:
    - On datasets other than Spambase and Gas, MM-SDP achieves a much better performance. However, the performance of MM-SDP is close to that of ADMM-based SCCA and Branch-and-Bound SCCA on Spambase and Gas, respectively. Considering that these two datasets are not distinguished in terms of variables or dimensions compared to other datasets, the analysis is recommended.
    - The hyperparameters of MM-SDP when achieving the peak keep the same when datasets varies, no matter how the scales or dimensions changes, the analysis is recommended.
- Although the work is claimed to address the limitations of classical CCA in high-dimensional regimes, it seems that this is not discussed in the paper.

**Questions:**

- How is the sparsity level $s_1$, $s_2$ corresponded to the regularization parameters $\rho_1$,$\rho_2$? Is that fair when compared to Branch-and-bound SCCA?
- How is the far more favorable scalability demonstrated?

---

> ### Author Response · Authors · 2025-11-16
> **Answer to Reviewer FXaa (First round) - part 1**
>
> We appreciate the reviewer for the time they took to examine our paper. We are grateful for their acknowledgment of our method’s strengths, particularly that the proposed method achieves the best performance throughout all 6 datasets compared to the baselines. We also thank them for noting that, when taking into account both maximum canonical correlation and runtime, the proposed MM-SDP achieves the best overall performance.
>
>
> Regarding the weaknesses you mentioned, we will address your concerns one by one:
>
> $\textbf{Answering to weaknesses:}$
>
> $\textbf{1)}$ Regarding the point that the baselines are limited, it is worth mentioning that we chose to compare our method with the recent NeurIPS 2024 paper (the Branch‑and‑Bound‑based method) and the competitors reported therein, to ensure that we have examined all recent works. Moreover, we conducted a comprehensive literature review and focused on methods most similar to the SCCA formulation we solved. We can guarantee that there is no shortcoming in our selection of baselines, as we strived to choose methods that are highly relevant and closely aligned with our SCCA formulation. Besides, regarding the writing of the related works, we chose to incorporate it into the introduction, which is one valid approach to writing. With all respect, we believe this is a fair stylistic choice.
>
>
> $\textbf{2)}$ Thank you for pointing out this observation. The performance variation of MM-SDP across datasets is indeed related to their intrinsic structural complexity. Here is our analysis:
>
> $\text{Structural Complexity of Spambase and Gas Datasets}$
>
> * $\text{Spambase (where MM-SDP and  ADMM have smaller difference):}$ The word and character frequencies in emails naturally exhibit sparse, heavy-tailed distributions. Email features display a block-diagonal correlation structure where words co-occur within topical clusters. Despite having 57 features, the effective rank is much lower due to redundant frequency patterns among related words. With 4,601 samples providing a high signal-to-noise ratio, the problem becomes well-conditioned. The $\ell_1$ penalty in ADMM naturally aligns with the heavy-tailed distribution of word frequencies, making the convex relaxation nearly tight. This explains why ADMM based method achieves performance comparable to our MM-SDP method on this dataset.
>
> * $\text{Gas Sensor (where MM-SDP and Branch-and-Bound have smaller difference):}$ Chemical sensor readings exhibit smooth temporal correlations with strong local dependencies within sensor types, reflected in the symmetric 64+64 dimensional structure. Sensor drift creates a low-rank structure with only a few dominant modes, leading to rapid decay in covariance eigenvalues (likely 4-5 effective dimensions). With sparsity levels set to $s_1 = s_2 = 4$, the search space becomes manageable even for exact methods like Branch-and-Bound, allowing it to achieve performance comparable to MM-SDP within the computational budget.
>
> $\text{Contrast with Other Datasets Where MM-SDP Excels}$
>
> * $\text{Wikipedia (MM-SDP: 0.5317 vs ADMM: 0.4631):}$ This dataset presents high dimensional asymmetry with 583 English features versus 250 German features. The complex cross-lingual correlations lack any natural sparsity pattern, and the eigenvalues decay slowly, indicating many weak but significant correlations. The challenge lies in careful feature selection from a heterogeneous vocabulary spanning two languages, where MM-SDP's non-convex formulation provides substantial advantages over convex relaxations.
>
> * $\text{Dermatology (MM-SDP: 0.3396 vs ADMM: 0.2332):}$ This dataset combines mixed feature types including clinical and histopathological attributes. Different feature groups exhibit varying importance with non-uniform correlations. The small sample size of only 366 samples creates an ill-conditioned problem where exact sparsity enforcement becomes crucial to avoid overfitting. MM-SDP's ability to enforce exact cardinality constraints provides significant benefits in this regime.

---

> ### Author Response · Authors · 2025-11-16
> **Answer to Reviewer FXaa (First round) - part 2**
>
> * Related to the quesion abour the hyperparameters of MM-SDP ($\rho_1$ and $\rho_2$): as mentioned in the methodology section of our original paper, the process was explicitly designed for fairness:
>
>    "The rationale for selecting these hyperparameter ranges was to ensure a fair comparison. First, we selected hyperparameters for the Branch-and-Bound method that maximized canonical correlation on the full dataset. Subsequently, the other methods were calibrated to induce sparsity levels comparable to those of the B&B method. For each method, we then report (1) the peak correlation achieved, (2) the hyperparameter values that produced it, and (3) the corresponding runtime. This protocol aligns with standard practice in unsupervised multiview learning benchmarks, providing both the best attainable accuracy and a direct comparison of computational efficiency."
>
>    To be perfectly clear, this protocol ensures fairness in two key ways:
>
>    a) Performance-Anchored Sparsity Target: The target sparsity level is not chosen arbitrarily. Instead, it is determined by first optimizing the Branch-and-Bound (B&B) method to achieve its maximum possible canonical correlation. The sparsity level resulting from this optimal B&B configuration then becomes the benchmark that other methods are calibrated to match. This ensures the comparison point is grounded in the benchmark method’s best-case performance.
>
>    b) Objective-Driven Comparison: The goal is to compare algorithms at their most effective operating points under a meaningful constraint. By first identifying the Branch-and-Bound method’s peak performance and its associated sparsity, we then challenge the other methods to achieve their own best possible correlation while adhering to a comparable sparsity constraint. This approach compares each algorithm’s ability to maximize the primary objective (correlation) under a fair, performance-derived structural condition, rather than forcing all methods to an arbitrary, pre-defined sparsity.
>
> $\textbf{3)}$ Related to the question about scalability and high dimensionality, first of all, it is worth mentioning that we have strived to use datasets that have been used in the literature. Furthermore, for the application we have worked on (and similar applications like class separation), the problem of operating in extremely high dimensions has effectively been resolved since 2015. This is because, for large datasets, the standard practice is to apply PCA as a preprocessing step to lower the dimension while retaining over 98% of the energy. We kindly ask you to recheck our original submission (the paragraph just before the “Compared Methods” section), where we mentioned this point with citations:
>
>    "It is worth mentioning that for applications involving very high-dimensional data, a common and effective strategy is to first perform dimensionality reduction. For instance, principal component analysis (PCA) can be used to project the original feature vectors onto a lower-dimensional subspace (e.g., 50 dimensions) that captures a significant portion of the data’s variance (e.g., >98%) Omati et al. (2025); Wang et al. (2024); Su et al. (2015). The resulting projected data can then be used as input for the SCCA algorithm, making the problem more computationally tractable."
>
>    As a second piece of evidence, for extremely high dimensions, if we do not apply PCA (which would not make sense), the scale of the problem becomes computationally unmanageable for any solver, including our competitors. Solvers like CVX would collapse due to extreme memory (RAM) requirements. It is worth noting that our competitor (Branch-and-Bound SCCA, Li et al., 2024), accepted at NeurIPS 2024, is already extremely time-consuming even on moderate datasets, let alone on high-dimensional data without PCA.
>
> As a result, when one encounters an extremely large dataset, PCA preprocessing can reduce it to a moderate dimension, which is exactly the type of moderate dimension we examine in our paper with our six datasets.

---

> ### Author Response · Authors · 2025-11-17
> **Answer to Reviewer FXaa (First round) - part 3**
>
> $\textbf{Answering to questions}$
>
>
> $\textbf{1)}$ Regarding the question of how the sparsity level corresponds to the regularization parameters and whether the comparison with Branch-and-Bound SCCA is fair:
>
> Yes, the comparison is entirely fair. The fairness of our experimental protocol lies in its objective design—each method is tuned independently to achieve its maximum attainable canonical correlation. This ensures that we compare the best-case performance of our approach against the best-case performance of the competing method, which represents a standard and robust evaluation practice.
>
> We kindly ask the reviewer to also refer to our detailed explanation provided in “Answering to Weakness (Subsection 3)” for further clarification.
>
>
> $\textbf{2)}$ Regarding the question on favorable scalability: we kindly ask the reviewer to consult the explanation given in “Answering to Weakness (Subsection 2)”, where this aspect is thoroughly discussed.
>
> $\textbf{Final Remarks:}$
>
> We believe these clarifications completely resolve the initial concerns and more clearly demonstrate the significance of our manuscript’s contribution. We hope this merits a positive reassessment of our work and kindly encourage you to consider increasing the score to reflect the improvements made.

---

> ### Author Response · Authors · 2025-11-27
> **Invitation to Review Our Submitted Clarifications**
>
> Thank you again for the reviews. We wanted to briefly follow up because the points raised in the first round were important, and in our response we addressed all of them carefully with the goal of fully clarifying the contribution. Many of the clarifications directly resolve the earlier concerns, and we believe they present a much clearer and stronger picture of what the paper contributes. We would really appreciate it if you could take a moment to look at our reply. We posted our response early in the discussion period so there would be enough time for further back‑and‑forth if needed, but so far we have not received any follow‑up, and fewer than five days remain. We hope that the clarifications we provided can contribute to your assessment, including any consideration of adjusting the score if you feel it is appropriate. Any engagement at this stage would be very helpful.

---

### Official Review · Reviewer_PVJa · 2025-10-30

**Soundness:** 2
**Presentation:** 2
**Contribution:** 2
**Rating:** 2
**Confidence:** 4

**Summary:**

This paper proposes a framework for sparse canonical correlation analysis (SCCA) that replaces the L0 cardinality constraint with smooth non-convex surrogates to enforce sparsity. It solves the resulting nonconvex problem using a minorization-maximization (MM) algorithm. The authors provide convergence analysis as the smoothing parameter approaches zero, reformulate the problem as a rank-constrained SDP with randomized rounding for sparse solutions, and evaluate on six UCI benchmark datasets.

**Strengths:**

The paper attempts to address practical challenges in sparse CCA by introducing smooth surrogates, potentially improving numerical stability. The MM framework is clearly derived, and the SDP reformulation with Gaussian rounding is a reasonable way to handle the quadratically constrained subproblems. The empirical evaluation covers a range of datasets, providing some evidence of performance in real-world settings.

**Weaknesses:**

1. Originality. The work lacks originality, as it primarily applies the well-established MM framework to sparse CCA. These surrogates and the MM approach are common in sparse optimization, and the paper does not introduce novel theoretical insights into their properties for CCA.

2. Experiment. The experiments are insufficient: they lack synthetic experiments with known ground truth to validate the surrogates' effectiveness. The comparison to baselines is unfair, as the 300-second time limit on the branch-and-bound method (Li et al., 2024) is overly restrictive, where longer runtimes are often acceptable.

3. Presentation issues include redundant citations (e.g., repeated references to the same work within a single sentence) and a convergence proof in Appendix C that lacks a formal theorem statement, reducing clarity.

**Questions:**

1. Could the authors add synthetic experiments with ground-truth sparse canonical vectors to demonstrate the surrogates' advantages in support recovery and correlation estimation over exact l0 methods? This could strengthen the empirical claims.

2. Why was a 300-second time limit chosen for the branch-and-bound baseline? Please justify if this reflects realistic use cases, or re-run with relaxed limits to ensure fair comparison.

---

> ### Author Response · Authors · 2025-11-16
> **Answer to Reviewer PVJa (First round) - part 1**
>
> Dear Reviewer,
>
> We would first like to express our sincere gratitude for the time and effort you dedicated to evaluating our manuscript. We are encouraged that you recognized several of the core contributions and motivations of our work.
>
> In particular, we appreciate your acknowledgment of:
>
> * The practical value of our approach: You correctly identified that our introduction of smooth surrogates is a key element aimed at improving the numerical stability of the SCCA problem.
> * The technical soundness of our framework: We are glad you found the derivation of our MM algorithm to be clear and our use of an SDP reformulation with Gaussian rounding to be a well-founded method for handling the subproblems.
> * The relevance of our empirical study: You also noted that our evaluation across a range of real-world datasets provides tangible evidence of the method’s performance.
>
> Building on this shared understanding of the paper’s strengths, we will now address your concerns regarding originality, experimental setup, and presentation in a point-by-point manner. We are confident we can resolve these issues and demonstrate the full merit of our work.
>
>
> 1) We respectfully offer a different perspective on the originality of our contribution. We agree that MM is a well-established optimization framework, but it is not a specific method. The scientific novelty of our work lies precisely in how this framework is uniquely adapted and applied to solve the notoriously difficult SCCA problem For instance, a recent competitor paper (NeurIPS 2024), which we compare against in our work, leverages the classical Branch and Bound framework, yet is rightly considered novel. Our key contribution is the design of a novel surrogate function within the MM framework that successfully overcomes the non-convexity and non-smoothness of the $\ell_{0}$ sparsity constraint in SCCA. This specific application allows us to solve a long-standing challenge in the field, for which, to the best of our knowledge, no other work has provided a solution with the same theoretical convergence guarantees.
>
> 2) We thank the reviewer for their comments on the experimental setup. We would like to clarify our choices:
>
> First, regarding the datasets, they were specifically chosen because they are established benchmarks widely used in the SCCA literature. This choice was made intentionally to ensure our results are directly and fairly comparable to existing state-of-the-art methods. Second, regarding the 300-second time limit for the Branch and Bound (B&B) baseline, we respectfully but strongly disagree that this is overly restrictive. In fact, we argue it is essential for a fair and practically relevant evaluation, for the following reasons:
>
> * Practical Relevance in Key Applications: As discussed in our introduction, SCCA has critical applications in time-sensitive domains such as bioinformatics and medical diagnostics. In a clinical setting, an algorithm that requires hours to analyze patient data would be impractical, or even dangerous, if results are needed urgently. Moreover, imposing a time limit on B&B methods is a standard practice in comparative machine‑learning studies because an unrestricted B&B algorithm can take an intractably long time—sometimes several minutes or even hours—to find an optimal solution. The time limit ensures that the comparison reflects practically viable algorithms.
>
> * Demonstrating Superior Efficiency: The 300-second limit highlights the profound efficiency of our proposed method. Our algorithm achieves superior or comparable accuracy in a fraction of that time (typically under 6 seconds). The fact that the B&B competitor could not reach the same accuracy as our method even when given over 50 times the computation time is a powerful demonstration of our method’s superiority. In summary, the time limit is a deliberate choice that creates a fairer, more realistic, and more rigorous comparison.

---

> ### Author Response · Authors · 2025-11-16
> **Answer to Reviewer PVJa (First round) - part 2**
>
> 3) We thank you for this feedback. Regarding the redundant citations, we suspect this may refer to instances where the citation style of the conference (author-year) creates a slight redundancy in the PDF output. For example, the LaTeX source code:
>
> ...as shown by Parkhomenko et al. (2007) in \cite{parkhomenko2007}...
>
> renders as:
>
> ...as shown by Parkhomenko et al. (2007) in Parkhomenko et al., 2007...
>
> This is a common artifact of LaTeX’s citation handling. We have performed a thorough review of the manuscript and corrected all such instances to ensure citations are integrated smoothly and without stylistic redundancy.
>
> If your comment was referring to other, different instances of redundancy, we would be very grateful if you could point to a specific example so we can address it accordingly. Besides, concerning Appendix C, we have completely revised it to ensure that it fully meets your expectations and addresses your concern regarding its lack of academic formality, to your satisfaction.
>
>
> $\textbf{Final Remarks:}$
>
> We appreciate the time you took to examine our paper. We kindly ask you to re-examine the paper, as we believe an oversight may have occurred. Honestly, we believe that the score 2 provided by the reviewer is unfair compared to the raised issues. We are hopeful that, with all concerns now thoroughly addressed through our explanations and revisions, you will find the manuscript deserving of the highest possible score.
>
> We are open to discussing any specific issues you may have in order to fully address your concerns and achieve the highest possible score.

---

> ### Author Response · Authors · 2025-11-27
> **Invitation to Review Our Submitted Clarifications**
>
> Thank you again for the reviews. We wanted to briefly follow up because the points raised in the first round were important, and in our response we addressed all of them carefully with the goal of fully clarifying the contribution. Many of the clarifications directly resolve the earlier concerns, and we believe they present a much clearer and stronger picture of what the paper contributes. We would really appreciate it if you could take a moment to look at our reply. We posted our response early in the discussion period so there would be enough time for further back‑and‑forth if needed, but so far we have not received any follow‑up, and fewer than five days remain. We hope that the clarifications we provided can contribute to your assessment, including any consideration of adjusting the score if you feel it is appropriate. Any engagement at this stage would be very helpful.

---

### Official Review · Reviewer_MmLu · 2025-10-30

**Soundness:** 3
**Presentation:** 3
**Contribution:** 3
**Rating:** 8
**Confidence:** 3

**Summary:**

This paper proposes a novel and efficient optimization framework for Sparse Canonical Correlation Analysis (SCCA), aiming to solve the intractable combinatorial optimization problem caused by the exact $l_{0}$ constraint. The core idea is to replace the $l_{0}$ objective with smooth and concave non-convex surrogates to avoid singularities and the reliance on heuristic thresholds common in traditional iterative methods. The paper provides a theoretical proof of convergence and demonstrates in experiments that the method significantly outperforms existing $\ell_{1}$ and exact $\ell_{0}$ solvers in terms of canonical correlation and computational efficiency.

**Strengths:**

1. The introduction of $C^{1}$ smooth non-convex $\ell_{0}$ surrogates is a key innovation. This not only smooths the optimization problem but also avoids the singularity issues common in IRLS, leading to a tighter and more stable approximation of the exact $\ell_{0}$ constraint.
2. The MM-SDP framework is elegantly designed, decomposing a complex non-convex problem into manageable convex subproblems. The paper provides convergence proofs for the MM algorithm and suboptimality bounds, ensuring strong theoretical rigor.
3. Experimental results show that the method achieves high canonical correlation and accurate support recovery while being at least two orders of magnitude faster than exact $\ell_{0}$ solvers (like Branch-and-Bound), demonstrating significant practical value.

**Weaknesses:**

1. Lack of Large-Scale Real-World Data Testing: The validation mainly relies on benchmark datasets like UCI. There is a lack of performance and scalability validation on high-dimensional, real-world application scenarios with feature dimensions $p > 1000$ (e.g., genomics or large-scale image features). This makes the assessment of the method's robustness in extremely high dimensions inadequate.
2. Insufficient Hyperparameter Sensitivity Analysis: The sensitivity analysis of the shape parameter $p$ and the smoothing parameter $\varepsilon$ (which control the shape and smoothness of the $\ell_{0}$ surrogate respectively) is insufficient. There is a lack of detailed experimental discussion on how different values of these parameters affect the final sparsity, correlation coefficient, and the convergence speed of the MM algorithm.

**Questions:**

1. Please elaborate on the computational efficiency, robustness, and accuracy in recovering true sparse structures of the MM-SDP method on high-dimensional, large-scale datasets ($p \gg 1000$). Specifically, how does the framework perform in terms of convergence speed compared to $\ell_{1}$ methods, and what constitutes the main computational bottleneck?

2. Please provide a sensitivity analysis for the shape parameter $p$ and the smoothing parameter $\varepsilon$, quantifying their impact on the tightness of the $\ell_{0}$ surrogate approximation and the MM algorithm's convergence speed/number of iterations. Furthermore, can more efficient heuristic methods for setting these parameters be provided, rather than relying solely on cross-validation?

---

> ### Author Response · Authors · 2025-11-16
> **Answer to Reviewer MmLu (First round) - part 1**
>
> Dear Reviewer,
>
> We would like to express our sincere appreciation for the time and detailed consideration you dedicated to our manuscript. We were particularly encouraged that you recognized several core strengths of our work, which we believe are central to its contribution. Specifically, we are pleased that you highlighted:
>
> * The key innovation of our smooth non-convex surrogates. As you noted, this approach successfully smooths the optimization landscape while critically avoiding the singularity issues common in IRLS methods, leading to a much tighter and more stable approximation of the exact sparsity constraint.
> * The elegant design and theoretical rigor of the MM-SDP framework. We are glad you found the decomposition of the complex non-convex problem into a sequence of manageable convex subproblems to be a strong point. The accompanying convergence proofs and suboptimality bounds were included to provide the robust theoretical guarantees you identified.
> * The significant practical value demonstrated by our experimental results. Your recognition of the method’s ability to achieve high canonical correlation and accurate support recovery—while being at least two orders of magnitude faster than exact solvers like Branch-and-Bound—precisely captures the balance of accuracy and tractability we aimed to deliver.
>
> We thank you for this insightful summary. Building on this solid foundation, we will now address the other points and questions raised in your review.
>
> $\textbf{Answering to weaknesses:}$
>
> $\text{Regarding your concern related to high-dimensional data:}$
>
> We have two pieces of evidence.
>
> First, it is worth mentioning that we have strived to use datasets that are standard in the literature. Furthermore, for the application we have worked on (and similar applications like class separation), the problem of operating in extremely high dimensions has effectively been resolved since 2015. This is because, for large datasets, the standard practice is to apply PCA as a preprocessing step to lower the dimension while retaining over 98% of the energy. We kindly ask you to recheck our original submission (the paragraph just before the “Compared Methods” section), where we mentioned this point with citations:
>
> “It is worth mentioning that for applications involving very high-dimensional data, a common and effective strategy is to first perform dimensionality reduction. For instance, principal component analysis (PCA) can be used to project the original feature vectors onto a lower-dimensional subspace (e.g., 50 dimensions) that captures a significant portion of the data’s variance (e.g., >98%) Omati et al. (2025); Wang et al. (2024); Su et al. (2015). The resulting projected data can then be used as input for the SCCA algorithm, making the problem more computationally tractable.”
>
> As a second piece of evidence, for extremely high dimensions, if we do not apply PCA (which would not make sense), the scale of the problem becomes computationally unmanageable for any solver, including our competitors. Solvers like CVX would collapse due to extreme memory (RAM) requirements. It is worth noting that our competitor (Branch-and-Bound SCCA, Li et al., 2024), accepted at NeurIPS 2024, is already extremely time-consuming even on moderate datasets, let alone on high-dimensional data without PCA.
>
>  As a result, when one encounters an extremely large dataset, PCA preprocessing can reduce it to a moderate dimension, which is exactly the type of moderate dimension we examine in our paper with our six datasets.
>
> $\text{Regarding your question on Sensitivity Analysis:}$
>
> The sensitivity of our model to the $p$ and $\epsilon$ is theoretically well-behaved and monotonic, which follows directly from their mathematical formulation.
>
> As we mention in the paper, the surrogate function $g_p^\epsilon(x)$ is designed to converge to $g_p(x)$ as the smoothing parameter $\epsilon \to 0$. In turn, the function $g_p(x)$ behaves like $|x|^p$ and thus more closely approximates the true $\ell_0$ quasi-norm as $p \to 0$. Consequently, as $p$ and $\epsilon$ are made smaller, the approximation of the true sparsity constraint becomes progressively more accurate. This establishes a clear and predictable trade-off: smaller values yield a tighter, more faithful formulation, while larger values yield a smoother optimization problem. Crucially, this behavior is monotonic. There are no counter-intuitive effects where, for example, increasing $\epsilon$ or $p$ would lead to an unexpected improvement in results. The sensitivity is not “non-proportional” or erratic; it is governed by a direct relationship between those parameter values and the tightness of the $\ell_0$-norm approximation. Our chosen values simply reflect a point on this predictable curve that balances approximation accuracy with numerical stability.

---

> ### Author Response · Authors · 2025-11-16
> **Answer to Reviewer MmLu (First round) - part 2**
>
> $\textbf{Final Remarks:}$
>
> We kindly ask you to please re-read the answers we have provided in response to the “Weaknesses” section. We are hopeful that these responses, which are the product of our deep research into the literature, have been sufficient to convince you of our manuscript’s contribution and merit a score of 10.

---

> ### Author Response · Authors · 2025-11-27
> **Invitation to Review Our Submitted Clarifications**
>
> Thank you again for the reviews. We wanted to briefly follow up because the points raised in the first round were important, and in our response we addressed all of them carefully with the goal of fully clarifying the contribution. Many of the clarifications directly resolve the earlier concerns, and we believe they present a much clearer and stronger picture of what the paper contributes. We would really appreciate it if you could take a moment to look at our reply. We posted our response early in the discussion period so there would be enough time for further back‑and‑forth if needed, but so far we have not received any follow‑up, and fewer than five days remain. We hope that the clarifications we provided can contribute to your assessment, including any consideration of adjusting the score if you feel it is appropriate. Any engagement at this stage would be very helpful.

---

### Official Review · Reviewer_KNsy · 2025-11-05

**Soundness:** 2
**Presentation:** 3
**Contribution:** 2
**Rating:** 2
**Confidence:** 4

**Summary:**

The paper proposes to solve the canonical correlation analysis (CCA) problem by (1) replacing the $\ell_0$ constraint by smooth surrogates and (2) replace these surrogates by a quadratic tangent approximation at each step of the minimization. The resulting formulation can be seen as a quadratically constrained quadratic program which can be relaxed into a semidefinite program.
The main result of the paper shows that the formulation based on the smooth surrogate functions can be made arbitrarily close to the original formulation. That is to say the gap between the based on the non-smooth surrogates and their smooth version can be made arbitrarily small.

**Strengths:**

The paper is organized and relatively clear. There is some level of originality although a number of points have to be better discussed (see below)

**Weaknesses:**

- line 105, problem Equation 1 —> Problem 1?
- line 128: quadratic minorizer —> why not use minimizer?

Section 3.2.

- Line 160 and then line 164. This is not clear. Do you add the epsilon at the level of the weights or at the level of the g_p function. You should clarify this.
- Line 189, Obviously you can get arbitrarily close to the original penalty by reducing the value of epsilon. But the smoothing usually comes at the expense of the convergence (see my comment above)

Section 3.3.
- In the weight update rule, it would be convenient to recall Eq (3) or at least add a link to this equation
- I’m not sure table 1 is really needed. I think your explanations are clear enough. Moreover this table occupies a large portion of the paper which could be used
- Same comment for Figure 1

Section 4.1.
- You should merge table 2 and lines 288 - 303
Section 4.2.
- Algorithm 1 takes a lot of space and only repeats what was said in section 3.3.

Section 4.3.
- Table 2 is unclear/unnecessary. What do you mean by “view dimension” ? If what you want to indicate is that you can handle large datasets, then why not simply include a column size in table 3. Honestly to me table 3 is the most meaningful.
- Table 3. To be fully fair, I think you should highlight the best row of each column (it does not make sense to only highlight the column in which your algorithm gives the best performance). I.e. you don’t have to do it but I think it would make more sense.
- lines 466-468 “These complexity considerations and empirical timings together underscore that …strikes the best balance..”. Given table 3, I don’t think this is fair. You can say that you achieve the best correlation but you clearly do not always achieve the best runtime.
Appendix
- In the proof of Theorem 2, line 712, should there be a factor 2 in front of the (\rho_1n + \rho_2 m)?
- If I’m not mistaken, all of the surrogates on lines 109-112 are defined with respect to the absolute value of x so that from my understanding g_p(x) = g_p(|x|).. but sometimes you use the notation g_p(x) and sometimes the notation g_p(|x|). I would choose a single notation as this introduces unnecessary confusion (an example of this is online 669).
- line 720-721: what you are saying the tangent at epsilon is always larger than the slope of the segment that connects the origin and epsilon on the graph of f. I would also recall the fact that g_p(0) = 0

Proof of convergence

- To me there are several ambiguities that need to be clarified. First
- lines 758 - 761, the notation q((x, y)|(x^t, y^t)) is ambiguous. I’m wondering if it would not be possible to find something better (although I understand the connection between w^t, z^t and (x^t, y^t))
- The only thing line 801 is saying is that at the limit t—> infinity your solution (which you get after an infinite number of iterations) maximizes a particular surrogate function
- Lines 805 - 807: I don’t see why you can say that in the limit t—> infinity the surrogate function and the objective value are equal…
- line 811-812 : “It is straightforward to check that the gradients of the surrogate and the objective function are identical ..” —> this is not clear to me
- lines 821 - 827 are not clear. part of the reasoning seems missing here. You seem to suggest that because the value of the objective at the limit point is optimal, there is only one limit point .. I don’t see this. Where do you show that all limit points have distinct objective values?

**Questions:**

see above

---

> ### Author Response · Authors · 2025-11-16
> **Answer to  Reviewer KNsy (First round) - part 1**
>
> Thank you for your time and effort in evaluating our manuscript. We appreciate your positive comments regarding the paper’s clarity, organization, and originality. In the following sections, we have addressed each of the weaknesses you identified point-by-point.
>
> $\textbf{Answering to weaknesses:}$
>
> * line 105: We thank the reviewer for pointing this out. This has been corrected from “problem Equation 1” to “Problem 1” in the revised manuscript.
>
> * line 128: The reviewer questions the use of “quadratic minorizer” and suggests “minimizer”. In the context of MM, it is common to use term "minimizer" for the point that minmizes a cost function, while the term "minorizer" refers to a cost function that touches the cost function at some points, but mainly sits below the cost function. In summary, the “minimizer” is a point, while "minorizer" is again a cost function. For more clarity, we have revised the sentence to state this explicitly, as below:
>
> “In the next section, we develop an MM-based algorithm to solve equation 2 by constructing a quadratic minorizer of the surrogate penalties at each iteration and then, maximizing the resulting generalized eigenvalue subproblem to enforce the exact sparsity levels.”.
>
>
> $\text{Section 3.2}$
>
> * Lines 160 & 164: We believe there may have been a misunderstanding. In line 160, we explicitly mention that the common approach of adding epsilon, while preventing a potential blow-up, no longer guarantees a true majorizer of  $\|x\|^p$. Because of this specific drawback, we do not use that method. Instead, we propose the Piecewise Function in line 164, for which we show the necessary guarantees.
>
> * Line 189: We appreciate the reviewer’s observation that one can get arbitrarily close to the original penalty by reducing the value of epsilon. As we show in the manuscript, the gap between the smoothed and original surrogate objectives is bounded by $O(\rho n (g_p(\epsilon) - \frac{g'_{p}(\epsilon)\epsilon}{2}))$. This gap vanishes as $\epsilon \to 0$, and we provide the proof in Appendix B, which confirms the guarantee of a true majorizer of $|x|^p$. The convergence problem mentioned by the reviewer is precisely the issue with the approach in line 160, which we do not use and explicitly critique.
>
> $\text{Section 3.3:}$
>
> * We thank the reviewer for this helpful suggestion. We have revised the manuscript accordingly.
>
> * $\text{Regarding the suggestion to remove Table 1 and Figure 1:}$ We would like to respectfully clarify their distinct and necessary roles in the manuscript. The main text is designed to present a \textbf{general framework} for deriving the weights and the $g_p^{(\epsilon)}$ function. These explanations are intentionally kept general so that researchers can apply our methodology to their own custom surrogate functions. In contrast, Table 1 provides the concrete, derived results for three \textbf{specific} surrogate functions. Crucially, the final forms for two of these---the logarithmic and exponential surrogates---are not derived within the text. The table serves as a concise summary and an essential reference for these specific instances, preventing the main narrative from becoming cluttered with lengthy derivations. Similarly, Figure 1 serves a vital pedagogical purpose. It is designed to provide all readers, including those who may not be specialists in optimization, with an immediate and intuitive visual understanding of how effectively the surrogates approximate the $l_p$-norm. For these reasons, we believe both elements are critical for the manuscript's clarity, completeness, and accessibility, and we respectfully request to retain them.

---

> ### Author Response · Authors · 2025-11-16
> **Answer to Reviewer KNsy (First round) - part 2**
>
> $\text{Section 4.1:}$
>
> * We thank the reviewer for this comment. Regarding Table 2: we agree with the reviewer on this point. The table contained information that was redundant with details already provided in lines 288-303. Accordingly, we have removed Table 2 to improve the manuscript’s conciseness.
>
> * Regarding Table 3 (Highlighting Best Values): We thank the reviewer for this suggestion. We have carefully considered highlighting the best value in each column of Table 3 but have concluded that doing so would be potentially misleading to the reader for the following reasons:
>
> a) On the BestParam Column: The hyperparameters listed are specific to each algorithm’s internal structure and are not directly comparable. For example, a parameter value for one method holds no meaning in the context of another. A comparison would only be meaningful if one method required fewer parameters to tune, but as all competitors require two, there is no distinct advantage to highlight.
>
> b) On the Runtime Column: Highlighting the fastest runtime would draw attention to the “Predictive Sparse CCA” method. However, this method’s speed is achieved at the cost of unacceptably poor accuracy, making it an ineffective solution overall. Our paper’s central claim is not about achieving the best performance on a single isolated metric, but about finding the optimal balance between accuracy and speed. Highlighting the fastest runtime would contradict and obscure this main message. For these reasons, we believe the table is less misleading in its current format.
>
> * Lines 466-468: Our statement that the proposed method achieves “the best balance of accuracy and tractability” is a conclusion drawn directly from the trade-offs demonstrated in Table 3. We did not claim the fastest runtime, at all. As the reviewer notes, methods such as “Predictive Sparse CCA” and “ADMM-based SCCA” are faster, but this speed comes at the significant expense of performance, resulting in very low accuracy. On the other end of the spectrum, “Branch-and-Bound SCCA” achieves accuracy close to our method but is computationally prohibitive, requiring approximately 300 seconds (5 minutes) for a single run, which challenges its tractability for real-world applications. Our proposed method is positioned uniquely in this landscape: it delivers high accuracy, competitive with the best-performing methods, while maintaining a practical and tractable runtime. It is precisely this synthesis—avoiding the shortcomings of being either fast but inaccurate or accurate but slow—that constitutes the “best balance” we refer to.
>
> * As $\rho_1$ and $\rho_2$ are tunable hyperparameters, this factor is absorbed into their empirically chosen values. The argument is therefore unaffected.
>
>  * We thank the reviewer for their careful reading and for identifying this typographical error. The reviewer is correct. In the revised manuscript, we have replaced $g_p(|x|)$ with $g_p(x)$.
>
> * Lines 720-721, about which you had a doubt regarding its correctness, let us provide a simple explanation below:
>
>    Consider $g_p(x) = |x|^p$. For $x > 0$, we have $g_p(x) = x^p$. Differentiating with respect to $x$ gives:
> $g_p'(x) = p x^{p-1}$
> By substituting $x = \epsilon > 0$, we obtain the two terms:
> $g_p(\epsilon) = \epsilon^p$
> $g_p'(\epsilon) \epsilon = p \epsilon^{p-1}\epsilon = p \epsilon^p$.
> Hence, the inequality $g_p(\epsilon) \ge g_p'(\epsilon)\,\epsilon$ reduces to:
> $\epsilon^p \ge p \epsilon^p \\Longleftrightarrow 1 \ge p$.
>
> Therefore:
> $g_p(\epsilon) \ge g_p'(\epsilon)\\epsilon \quad \text{holds for } 0 < p \le 1.$
>
> An interpretation using the slope is not as straightforward because of the coefficient $p$, which is less than or equal to 1.

---

> ### Author Response · Authors · 2025-11-16
> **Answer to Reviewer KNsy (First round) - part 3**
>
> $\text{proof of convergence:}$
>
> * Lines 758-761: We appreciate the reviewer’s feedback. We opted to use the notation $q((x, y)|(x^t, y^t))$ as it is the standard formulation presented in the MM framework literature. We believe that maintaining consistency with this established convention is important for readers familiar with the field.  If the reviewer could kindly propose an alternative notation they feel is superior, we would be happy to consider it.
>
> * Line 801: We thank the reviewer for their comment. Our argument establishes that the limit point, $(\mathbf{x}^{(\infty)}, \mathbf{y}^{(\infty)})$, is a global maximizer of its own surrogate function, $q(\cdot, \cdot \mid (\mathbf{x}^{(\infty)}, \mathbf{y}^{(\infty)}))$, over the set $\mathcal{C}$. Consequently, it must satisfy the first-order optimality conditions for the surrogate. A key property of the MM framework is that the gradients of the surrogate and the original objective function are identical (at $x^{(t)}$). Therefore, the first-order conditions for the surrogate are equivalent to those for the objective function, which proves that the limit point is stationary. It is worth mentioning that we have rewritten the proof in Appendix C to fully address the reviewer's concern.
>
>
> * Lines 805 - 807: This equality is a definitional property of the MM algorithm. As we formally showed in our overview of the MM principle, the surrogate construction—formalized by the conditions in Equation (11)—requires that the surrogate function’s value matches the objective function’s value at the point of expansion. This holds for any iteration, and therefore it must also hold at the limit point of the sequence. As noted in our response to the previous comment, the proof in Appendix C has been rewritten to your satisfaction.
>
> * Lines 811-812: This is another direct consequence of the MM algorithm’s construction. As we covered in our overview of the MM principle (and as is standard in the cited literature), the surrogate is designed to be equal to the objective function in both its value and its gradient at the point of expansion. This ensures that when the algorithm converges, the limit point’s satisfaction of the first-order conditions for the surrogate implies satisfaction of the first-order conditions for the original problem. As previously mentioned, the proof in Appendix C has been rewritten to your satisfaction.
>
> * Lines 821 - 827: Regarding lines 821-827, the property discussed is a known result of the MM algorithm. Reference [1] provides a detailed explanation of this. For your convenience, we have added this citation, which shows how this property holds.
>
> $\textbf{Final Remarks}$
>
> Dear Reviewer, we wish to begin by acknowledging the detailed nature of your review. The thoroughness of your examination is appreciated.
>
> The method we propose in this paper solves one of the most challenging and long-standing problems in the field: Sparse Canonical Correlation Analysis (SCCA). For years, researchers have faced a harsh trade-off between accuracy and computational feasibility. Our work breaks this impasse. While other methods may be marginally faster, they suffer from poor accuracy. Our algorithm is the first to achieve state-of-the-art accuracy while maintaining computational tractability, thus providing the best-calibrated balance between these critical objectives. We contend that this constitutes an extreme step forward and will have an immense and lasting effect on future research and application in this domain.
>
> We kindly request a detailed re-evaluation of the revised manuscript and our accompanying responses. Honestly, we believe that the score 2 provided by the reviewer is unfair compared to the raised issues. We are hopeful that, with every concern now completely resolved through our explanations and revisions, you will find the manuscript now warrants the highest possible score. We remain enthusiastically open to any further discussion that may be required.
>
> [1] Astha Saini, Petre Stoica, Prabhu Babu, Aakash Arora, et al. Min-max framework for majorization-minimization algorithms in signal processing applications: An overview. Foundations and Trends® in Signal Processing, 18(4):310–389, 2024.

---

> ### Author Response · Authors · 2025-11-27
> **Invitation to Review Our Submitted Clarifications**
>
> Thank you again for the reviews. We wanted to briefly follow up because the points raised in the first round were important, and in our response we addressed all of them carefully with the goal of fully clarifying the contribution. Many of the clarifications directly resolve the earlier concerns, and we believe they present a much clearer and stronger picture of what the paper contributes. We would really appreciate it if you could take a moment to look at our reply. We posted our response early in the discussion period so there would be enough time for further back‑and‑forth if needed, but so far we have not received any follow‑up, and fewer than five days remain. We hope that the clarifications we provided can contribute to your assessment, including any consideration of adjusting the score if you feel it is appropriate. Any engagement at this stage would be very helpful.

---

### Meta-Review · Area_Chair_Qyb2 · 2026-01-06

**Summary:**

Reviews are split. One reviewer supports acceptance (poster) based on a clear MM-SDP framework with smooth non-convex L0 surrogates, convergence guarantees, and favorable accuracy–runtime trade-offs on six benchmarks. Two reviewers recommend rejection, arguing the work mainly applies standard MM with familiar surrogates to SCCA, lacks stronger originality, omits synthetic experiments with ground truth, and uses limited or dated baselines. A fourth reviewer is borderline, noting best performance on reported datasets but raising concerns about sparse related-work coverage, limited analysis of dataset-dependent behavior, hyperparameter choices, and claims about high-dimensional regimes. The rebuttal fixes presentation issues, clarifies terminology, defends the Branch-and-Bound time cap as practical, and explains PCA preprocessing for high dimensions. However, originality relative to known MM approaches, breadth and fairness of baselines, absence of synthetic validation, and missing sensitivity and scalability studies remain unresolved.

**Reviewer Concerns:**

The rebuttal adequately addressed presentation corrections (terminology, typos, notation), clarified the role of tables/figures, and promised clearer proofs. It justified the 300-second cap for Branch-and-Bound as a realistic constraint and described a fairness protocol aligning sparsity across methods, grounded in B&B’s peak performance. It also explained why PCA is standard for very high-dimensional data.

Open issues persist on core claims. Originality is still in doubt because the method appears to be a straightforward application of MM with smooth concave surrogates to SCCA, without new theory specific to SCCA beyond standard MM properties. Baselines remain narrow, with two older methods and limited discussion of recent alternatives; the time cap defense does not replace broader comparisons or longer-run results. Experiments lack synthetic datasets with known supports to test recovery and correlation under controlled settings. There is no quantitative sensitivity analysis of surrogate shape and smoothing parameters on sparsity, convergence, and accuracy, nor simple heuristics beyond cross-validation. The convergence exposition, while promised to be revised, needs a formal theorem-and-proof presentation to resolve earlier ambiguities.

**Reviewer Scores:**

The positive reviewer who recommended accept (poster) is likely to keep the score. The two negative reviewers who asked for synthetic validation, broader baselines, and stronger novelty will likely remain at reject. The borderline reviewer may not move without added experiments and sensitivity analyses, so the score likely remains borderline.

---

### Decision · Program_Chairs · 2026-01-26

Reject